



# Long-term Time-series of Arctic Tropospheric BrO derived from UV-VIS Satellite Remote Sensing and its Relation to First Year Sea Ice

Ilias Bougoudis[1], Anne-Marlene Blechschmidt[1], Andreas Richter[1], Sora Seo[1], John Philip Burrows[1], Nicolas Theys[2], Annette Rinke[3]

[1]Institute of Environmental Physics, University of Bremen, Bremen, 28359, Germany
[2]Belgian Institute for Space Aeronomy (IASB-BIRA), Brussels, Belgium
[3]Alfred Wegener Institute Helmholtz Centre for Polar and Marine Research, Potsdam, Germany

*Correspondence to*: Ilias Bougoudis (ibougoudis@iup.physik.uni-bremen.de)

**Abstract.** Arctic Amplification describes the rapid increase of the air temperature in the past three decades in the Arctic, which impacts on physicochemical conditions, the ecosystem and biogeochemistry. Every polar spring, the BrO explosion, a series of chemical reactions that release bromine molecules to the troposphere occurs over sea ice covered regions. This autocatalytic mechanism depletes boundary layer and tropospheric ozone, thereby changes the oxidizing capacity of the atmosphere and facilitates the deposition of metals (e.g. Hg). In this study, we present a 22 year consolidated and consistent tropospheric BrO dataset, derived from four different UV-VIS satellite instruments and investigate the BrO evolution under the impact of Arctic Amplification. The retrieval data products from the different sensors are compared during periods of overlap and show good agreement. By studying the sensor merged time-series of tropospheric BrO vertical column densities, we find an increase in the magnitude of BrO explosion events under the impact of Arctic Amplification with an upward trend of about 1.5% per year. Furthermore, the areas where BrO plumes frequently appear have changed, extending over larger regions in the Arctic during more recent years. Comparison to sea ice age data suggests that the reported changes in tropospheric BrO are linked in a complex way to the increase of first-year ice extent in the Arctic.

## 1 Introduction

Arctic temperature has risen at twice the rate of the global mean over the past three decades. This is called Arctic Amplification (Serreze and Barry, 2011) and our understanding of the processes leading to this phenomenon is inadequate. Important consequences of the rapidly increasing temperature in the Arctic are the loss of sea ice, the reduction of the sea ice extent (Stroeve et al., 2012) and the increasing rate of loss of the Greenland ice cap (Mouginot et al., 2019). Both the maximum and the minimum yearly sea ice extent began to be noticeably smaller over a decade ago. The minimum sea ice extent, which occurs usually in September, reached its record low in 2012 (Yang and Magnusdottir, 2018). The yearly maximum sea ice extent, which occurs every March, shrinks at a significant rate (Serreze and Meier, 2018). The thickness of





sea ice has declined dramatically in recent years as the portion of thick multi-year ice decreases (Richter-Menge et al., 2017). Sea ice is replaced by open ocean, which being darker, reduces surface reflectivity in the Arctic. As a result, more of the incoming solar radiation is absorbed by the ocean and its biosphere (e.g. phytoplankton). Consequently, temperature of the ocean and the air in the boundary layer increases, creating a positive feedback loop. This is one of the most pronounced

effects associated with Arctic Amplification (Hansen et al., 1997; Kirtman et al., 2013). These evolving conditions impact on many other physical, chemical and biological processes, as well as the eco system in the Arctic.

       Bromine monoxide (BrO) plays a significant role in the atmospheric chemistry of the Arctic. During polar springtime, episodes of strongly enhanced amounts of BrO have been observed in the boundary layer (Barrie and Platt, 1997). The formation of these intense plumes of BrO results in tropospheric ozone depletion. Tropospheric ozone depletion and the link

to halogen chemistry events was first discovered 30 years ago (Barrie et al., 1988) and has been the subject of many studies and campaigns over the past three decades (Toohey et al., 1990; Tuckermann et al., 1997). The release of reactive halogens and the decrease in $O_3$ impacts on the oxidizing capacity of the troposphere. The photolysis of $O_3$ in the UV-B leads to the formation of the most important tropospheric oxidising agent, the hydroxyl radical, OH. While in plumes of tropospheric BrO, the oxidising agents $O_3$ and OH are reduced, the reactions of BrO play also other important roles in atmospheric

chemistry. For example, BrO efficiently reacts with elemental mercury. This oxidation initiates a process whereby deposition of mercury to snow and ice increases. This results in Hg entering the food chain (Lu et al., 2001; Schroeder et al., 1998). The rapid and sudden appearance of BrO plumes over the Polar Regions has been called the "bromine explosion" (Barrie and Platt, 1997; Platt and Lehrer, 1997). It is explained by an autocatalytic multiphase chemical cycle, which occurs on cold saline surfaces (Fan and Jacob, 1992; Sander and Crutzen, 1996). The triggering mechanism for the release and the

production of gas phase plumes of BrO remains a matter of scientific controversy (Jones et al., 2009). However, there is the general consensus that the potential sources are (a) rich in sea salts and relatively cold (conditions occurring in potential frost flowers regions; Rankin et al., 2002; Kaleschke et al., 2004; Sander et al., 2006), (b) surfaces covered with liquid or frozen brine (Sander et al., 2006), (c) associated with blowing snow (Yang et al., 2008; Blechschmidt et al., 2016; Frey et al., 2019), (d) surface snow packs (Pratt et al., 2013; Peterson et al., 2018) and young sea ice regions (Wagner et al., 2001; Simpson et

al., 2007; Peterson et al., 2016). A pH lower than 6.5 is required for efficient bromine activation (Fickert et al., 1999; Halfacre et al., 2019).

       Organic sources of BrO, such as oceanic bromocarbons, have also been discussed in the literature (Salawitch et al., 2006, and references therein) but the relatively long lifetimes result in a slow release of Br throughout the Arctic troposphere and are currently not considered to explain observations of the bromine explosions. Starting with molecular bromine in the gas

phase, the sequence of autocatalytic chain reactions describing the bromine explosion reactions can be written in its simplest form as:

$Br_2 + h\nu$ (350 nm < $\lambda$ < 500 nm) → 2Br                    (R1)





$$2(Br + O_3 \rightarrow BrO + O_2) \tag{R2}$$

$$2(BrO + HO_2 \rightarrow HOBr + O_2) \tag{R3}$$

$$2(HOBr_{(g)} = HOBr_{(aq)}) \tag{R4}$$

$$2(HOBr_{(aq)} + Br^-_{(aq)} + H^+_{(aq)} \rightarrow Br_{2(g)} + H_2O_{(aq)}) \tag{R5}$$

Net: $2O_3 + 2HO_2 + 2Br^-_{(aq)} + 2\ H^+_{(aq)} \rightarrow 4O_2 + 2H_2O + Br_2$

In short, the autocatalytic multiphase chain reaction releases molecular bromine, $Br_2$, to be photolysed (R1). The resulting bromine atoms remove rapidly tropospheric $O_3$ (R2). The resultant BrO reacts with $HO_2$ to form HOBr (R3). This enters the aqueous phase or quasi liquid layers on cold brine or snow and ice, where it reacts with halogen ions to release $Br_2$ to the atmosphere (R5). The efficiency of such a chain reaction depends on the chain length in the atmosphere. This depends on the

relative rate of chain propagation and chain termination reactions of the chain carriers i.e. the Br and BrO. The bromine explosion slows through the depletion of $O_3$ in the air mass, or through reactions of Br or BrO with formaldehyde or $NO_2$:

$$Br + HCHO \rightarrow HBr + HCO \tag{R6}$$

$$BrO + NO_2 + M \rightarrow BrONO_2 + M \tag{R7}$$

$$BrONO_{2(g)} = BrONO_{2(aq)} \tag{R8}$$

There are also cycles involving chlorine ions, also initiating a catalytic loss. The involved reactions are explained in more detail elsewhere (e.g. Simpson et al., 2007; Sander et al., 2006).

It was shown that BrO plumes can be transported far from their initial formation areas, due to release from snow packs (Peterson et al., 2018) and blowing snow (Giordano et al., 2018) favored by high wind speeds associated with cyclones (Begoin et al., 2010; Zhao et al., 2015; Blechschmidt et al., 2016. Peterson et al. (2017) studied the vertical transport of BrO

and it's recycling on aerosol particles, stating that BrO can be sustained, reformed and transferred on aerosols. Using a qualitative model, Jones et al. (2009) have shown, that both stable boundary layer, very low near surface wind speed conditions, as well as unstable boundary layer, high wind speed conditions increase the number of reactants in the air and hence favor the bromine explosion.

The Polar Regions are some of the most remote and hostile places on the planet. Consequently, satellite remote sensing is

a unique method to study bromine chemistry in the Arctic. Richter et al. (1998) and Wagner and Platt (1998) were the first studies on satellite observations of BrO plumes in Polar regions, using the GOME instrument (Richter et al., 1998; Wagner and Platt, 1998). Based on observations from the same instrument, Hollwedel et al. (2003) derived a six year time-series of Arctic and Antarctic total vertical column densities (VCDs) of BrO. This was the first scientific effort to study the evolution of BrO in the Polar regions. The transport of BrO plumes, which represents a photo-stationary state with production and loss

processes and their capability of depleting ozone, far away from the initial release area was studied with satellite remote sensing (Ridley et al., 2007; Begoin et al., 2010). The relationship between BrO release and young sea ice was also discussed (Wagner et al., 2001). Van Roozendael et al. (2004) and Jacobi et al. (2006), used SCIAMACHY observations for Arctic BrO and compared them to GOME data, Theys et al. (2011) and Sihler et al. (2012) used GOME-2 for the same purpose.



Seo et al. (2019a) presented the first BrO retrievals from the TROPOMI instrument. Studies have also used satellite remote sensing, to link them to their sources and triggering meteorological conditions, in order to better understand this complex and significant phenomenon. For instance, Choi et al. (2018), Begoin et al. (2010), Toyota et al. (2011) and Jones et al. (2009) investigated links between tropospheric BrO and blowing snow. Blechschmidt et al. (2016) investigated a BrO explosion

event using GOME-2A and associated its long lifetime with continuous release of bromine molecules from blowing snow along the front of a polar cyclone.

Major changes in meteorological parameters, e.g. increasing air temperature (Serreze and Barry, 2011), decreasing mean sea level pressure over northeastern America and increasing pressure over Eurasia (Ogawa et al., 2018; McCusker et al., 2016), increase in cyclone frequency and intensity (Akperov et al., 2019), stronger surface winds (Mioduszewski et al., 2018) and

changes in sea ice conditions (e.g. reduced sea ice extent; Stroeve et al., 2012), increased first year sea ice fraction and therefore decreased sea ice thickness (Richter-Menge et al., 2017)) occur due to Arctic Amplification. It is therefore likely that the intensity, frequency and spatial distribution of bromine explosions in the Arctic are changing. Consequently, the objectives of this study are a) to derive the first consolidated and consistent long-term Arctic tropospheric BrO dataset using satellite remote sensing and b) to investigate and understand changes or trends in BrO and the relation to sea ice based on the

new dataset. As described above, an earlier effort was carried out by Hollwedel et al. (2003), but they did not extract the tropospheric BrO column from their 6 year GOME dataset. Moreover, Choi et al., (2018) used an 11 year Arctic tropospheric BrO dataset from the OMI instrument,, and investigated the link to young sea ice and sea salt aerosols released from blowing snow. However, the OMI sensor suffers from instrumental degradation, known as row anomaly (Class et al., 2010). In our study, we use measurements from four ultraviolet - visible satellite instruments to derive tropospheric VCDs of

BrO over the Arctic, covering a time-span of 22 years, starting with GOME observations in 1996 and ending with GOME-2A and GOME-2B in 2017. We first retrieve BrO slant column densities (SCDs) for each instrument and then separate tropospheric VCDs from stratospheric BrO amounts.

This paper is structured as follows: in section 2, a short technical description of the instruments used in this study is presented. In section 3, the methods applied for retrieving geometric and tropospheric BrO columns from the satellite sensors

and the additional datasets used for stratospheric separation and sea ice flagging of tropospheric BrO are introduced. In section 4, an analysis results of the long-term data is presented, including time-series and maps of tropospheric BrO VCDs and comparisons with sea ice age and a statistical analysis of possible trends of tropospheric BrO VCDs. The paper ends with a summary and conclusions (section 5).

## 2 Instruments

The technical attributes of the satellite sensors that are used, together with information about instrumental degradation are described in the following. A short summary of characteristics of the sensors is given in Table 1.



## 2.1 GOME

The GOME (Global Ozone Monitoring Experiment) (Burrows et al., 1999) instrument was launched in 1995 on ERS-2. It was a nadir viewing scanning spectrometer, observing solar back-scattered radiation upwelling from the Earth's surface and atmosphere. It measured continuously in four spectral channels from 240 to 790 nm, with a spectral resolution between 0.2

and 0.4 nm. For 27 days per month, the spatial resolution of the forward scans was 40x320 $km^2$, and the swath 960 km. For approximately 3 days per month, the instrument operated in narrow swath mode with a swath width of 240 km and ground scenes having a spatial resolution of 40x80 $km^2$. ERS-2 was in a sun synchronous orbit with a 10:30 local time equator overpass in the descending node. GOME was the first satellite instrument, whose measurements of the ultraviolet and visible upwelling radiance were appropriate for the retrieval of many key trace gases (Burrows et al., 1999). The instrument was

launched in July 1995 and lost its global coverage in 2003, due to data rate limitation (the onboard storage capability of the instrument was disabled and as a result data could only be transmitted directly to ground stations) (Bracher et al., 2005).

## 2.2 SCIAMACHY

SCIAMACHY (SCanning Imaging Absorption SpectroMeter for Atmospheric CHartographY) (Bovensmann et al., 1999) was a satellite spectrometer onboard Envisat. It was launched into space in 2002. The main advantage of SCIAMACHY

compared to GOME was its broader spectral coverage, ranging from 210 nm to 2380 nm, allowing the observation of many trace gases in the near infrared and short wave infrared spectral regions. SCIAMACHY's spectral resolution was 0.2 nm to 0.5 nm, and its spatial resolution in the spectral region used for BrO retrievals was 30x60 $km^2$, with a 960 km swath width in nadir. The overpass of Envisat over the equator was at 10:00 local time. SCIAMACHY observed the Earth in nadir, limb and occultation geometries, providing a wealth of trace gas data on the atmosphere from the surface to the upper atmosphere

(Bovensmann et al., 1999). In April 2012, Envisat lost contact with the ground station and as a result, the mission had to be terminated.

## 2.3 GOME-2

The series of GOME-2 (Global Ozone Monitoring Experiment–2) (Callies et al., 2000) instruments were developed as the successors of GOME. There are currently three instruments in orbit, one launched onboard Metop-A in 2006, one on Metop-

B in 2012, and one on Metop-C in 2018. Here, we use data from GOME-2A and GOME-2B. All GOME-2 instruments share the same attributes and sense the Earth's backscattered radiance and extraterrestrial solar irradiance in the ultraviolet and visible part of the spectrum (240 nm to 790 nm). They have a spectral resolution between 0.2 nm and 0.4 nm, while the footprint size is 80x40 $km^2$ and a much wider swath (1920 km) than the previous instruments. GOME-2A changed its swath to 960 km and footprint to 40x40 $km^2$ in June 2013. The GOME-2 instruments are crossing the equator at 09:30 local time

(Callies et al., 2000). All three instruments are currently in operation.



**Table 1: Attributes of the satellite instruments used in this study**

| Instrument | Platform | Period | Footprint | Equatorial Overpass | Swath |
|---|---|---|---|---|---|
| **GOME** | ERS-2 | 1996 – 2003 | 320x40 km$^2$ | 10.30 LT | 960 km |
| **SCIAMACHY** | Envisat | 2002 – 2012 | 30x60 km$^2$ | 10.00 LT | 960 km |
| **GOME-2A** | MetOp – A | 2007 – present | 80x40 km$^2$ <br> 40X40 km$^2$ (since June 2013) | 09.30 LT | 1920 km <br> 960 km (since June 2013) |
| **GOME-2B** | MetOp – B | 2012 – present | 80x40 km$^2$ | 09.30 LT | 1920 km |

## 2.4 Instrumental Degradation

Many space borne optical instruments suffer from a decrease in throughput in the ultraviolet spectral region, which arises from the deposition of absorbing layers on the optical surfaces such as mirror lenses or gratings. This results in a variety of
effects such as loss of throughput and changing etalons in the instrument. The quality of the retrieved BrO data, which is produced from the weak absorption signal in the UV region, is influenced by these degradations. The GOME, SCIAMACHY and GOME-2 teams identify any degradation in the reflectances and correct where appropriate and possible. For example organic compounds and water, emitted by the spacecraft, are photochemically transformed by UV-B and vacuum UV from the sun and most likely form long changing polymers, which have low vapour pressure and are absorbed on the mirrors. In
this context, Snel (2000), showed that the GOME sensor experienced degradation in all wavelength regions but in particular in the UV. Krijger et al. (2007), compared the degradation of GOME to SCIAMACHY with respect to the reflectivity. Dikty et al. (2011), investigated the impact of GOME-2A throughput loss on various time-series of trace gases. Most of the major degradation effects on the sensors are identified and documented in literature and where possible accounted for (Munro et al., 2016; Garcia et al., 2016).

## 3 Methods

In this section, the methodology to retrieve BrO SCDS from the satellite measurements and to obtain tropospheric BrO VCDs and datasets used in this study are described.

### 3.1 Retrieval of BrO slant column densities

Similar to previous studies on atmospheric BrO from hyperspectral satellite remote sensing observations in the solar spectral
regions, the retrieval algorithm is based on differential optical absorption spectroscopy (DOAS) (Platt and Perner, 1983; Burrows et al., 2011). DOAS is an application of the Beer and Lambert law, which describes the attenuation of electromagnetic radiation in a medium:



$$I = I_o \, e^{-\int(\sum_{j=1}^{J} \sigma_j(\lambda)\rho_j)ds}$$
Eq. (1),

where I is the measured intensity of the electromagnetic radiation, $I_o$ is the initial intensity, J is the total number of trace gases absorbing, j denotes a particular trace gas (e.g. BrO), $\sigma(\lambda)$ is the cross section of the absorber at wavelength $\lambda$, $\rho$ the density of the trace gas and s the length of the light path.

The main idea of the DOAS method is, that the atmospheric amount of the trace gases of interest can be retrieved using their characteristic spectral fingerprints. This is done by separating the extinction signal into a low frequency and a high frequency part. The low frequency part is treated as a closure term and is fitted by a low order polynomial. The higher frequency term contains the absorption structures of the trace gases. The final output of the retrieval is the slant column density of the trace gas, i.e. the density of the trace gas, integrated along the light path (Platt and Stutz, 2008):

$$SCD_j = \int \rho_j(s)ds$$
Eq. (2),

For cloud free conditions, the radiance upwelling at the top of the atmosphere is described at a given wavelength by:

$$I(\lambda) = cI_o(\lambda) \, e^{-\int \sum_{j=1}^{J} \{\sigma_j SCD_j - \sigma Ray(\lambda)SCD(Ray) - \sigma Mie(\lambda)SCD(Mie)\}ds}$$
Eq. (3),

where c is the scattering efficiency, $SCD_j$ the slant column density of the gas with index j, $\sigma Ray(\lambda)$ and $\sigma Mie(\lambda)$ are the absorption cross sections of Rayleigh scatterers (e.g. primarily air molecules, molecular nitrogen $N_2$ and oxygen, $O_2$) and

Mie scatterers (e.g. aerosol particles), SCD(Ray) and SCD(Mie) are the corresponding slant columns of Rayleigh and Mie scatterers. The Rayleigh and Mie terms in Equation 2 are the low frequency broad band features and they can be approximated by a polynomial:

$$\ln \frac{I_o(\lambda)}{I(\lambda)} = \sum_j \{\sigma_j(\lambda)SCD_j(\lambda)\} - \sum_\rho \{\sigma_\rho \lambda^\rho\}$$
Eq. (4)

To retrieve accurate BrO SCDs, an optimal selection of the spectral window, which maximises the information content

with respect to the BrO absorption, and the selection of corresponding cross sections of other trace gases absorbing in the same spectral window is the first step. We chose to use temperature dependent cross sections of ozone (dominant absorber in the UV) by Serdyuchenko et al. (2014) at 223 and 243 Kelvin and a BrO cross section (Fleischmann et al., 2004) at 223 Kelvin. In addition, a pseudo cross section is used for simulating the filling-in of Fraunhofer lines by Raman scattering known as the Ring effect (Vountas et al., 1998), and another pseudo cross section, which deals with the issue of poor spectral

sampling (Chance et al., 2005), was added for all instruments. The high spectral resolution absorption cross sections were convolved by the slit function of each instrument. The reason for using these cross sections is that our study is dedicated to BrO released in the Arctic region. Tests were made for different combinations of trace gases. It was found that the omission of the explicit fitting of $NO_2$ and $SO_2$ has a minimal impact on the fit of the BrO SCD: the differences between with and without $NO_2$ or $SO_2$ in the selected spectral ranges were typically less than 1% of the BrO SCD. Therefore, only cross

sections of the dominant absorber $O_3$ in this region, and of the trace gas of interest itself, BrO were added. Also, a four degree polynomial was used, as the differences between fourth and fifth order polynomials were small on the BrO SCDs, while the quality of the fit improved greatly from a third to fourth degree polynomial.



As BrO is a relatively weak absorber, small changes in the input parameters, and especially of the fitting window of the retrieval, can lead to large changes in the quality of the fit. Although there are many good practices developed for the DOAS fitting window selection for an absorber (e.g. it must include at least two absorption peaks from the trace gas of interest, no large Fraunhofer lines, small interference from other species etc.), there is no precise and agreed methodology to determine the optimal selection. As a result, different spectral fitting windows for BrO retrievals have been used in previous studies. For example, Richter et al., (1998), used a 345 - 359nm wavelength region for GOME, Afe et al., (2004), used a 336 - 347 nm fitting window for SCIAMACHY, while Theys et al., (2009) used a 336 to 359 nm fitting window for GOME-2A. Each of the sensors has slightly different instrumental characteristics, and each of them shows different degradation behaviour. Consequently, we chose different fitting windows for each of the sensors that result in a) a low root mean squared error (RMS) of the fit, b) a reduced trend of BrO SCDs over a clear Pacific reference region, where no strong trend in BrO is expected and c) good agreement between retrievals from the different sensors for periods with more than one instrument in operation. The RMS of the fit is defined as:

$$\sum_{j=1}^{J} = \sqrt{\frac{\left(\ln\left(\frac{I_0}{I}\right) - (SCD_j\,\sigma(\lambda)_j) - P(\lambda)\right)^2}{N}}$$  Eq. (5),

where N is the number of wavelengths and $P(\lambda)$ the low order polynomial. The RMS has been evaluated for the region of interest (i.e. the Arctic from $70.0^o$ to $85.0^o$ latitude, $-180^oE$ to $180^oE$ longitudes) and for the Pacific reference region ($-50.0^oS$ to $10.0^oN$ latitude and $235.0^oE$ to $270.0^oE$ longitude). In Figure 1, SCDs of BrO over the Pacific reference region and the RMS of the fit for the Arctic and Pacific regions are shown.

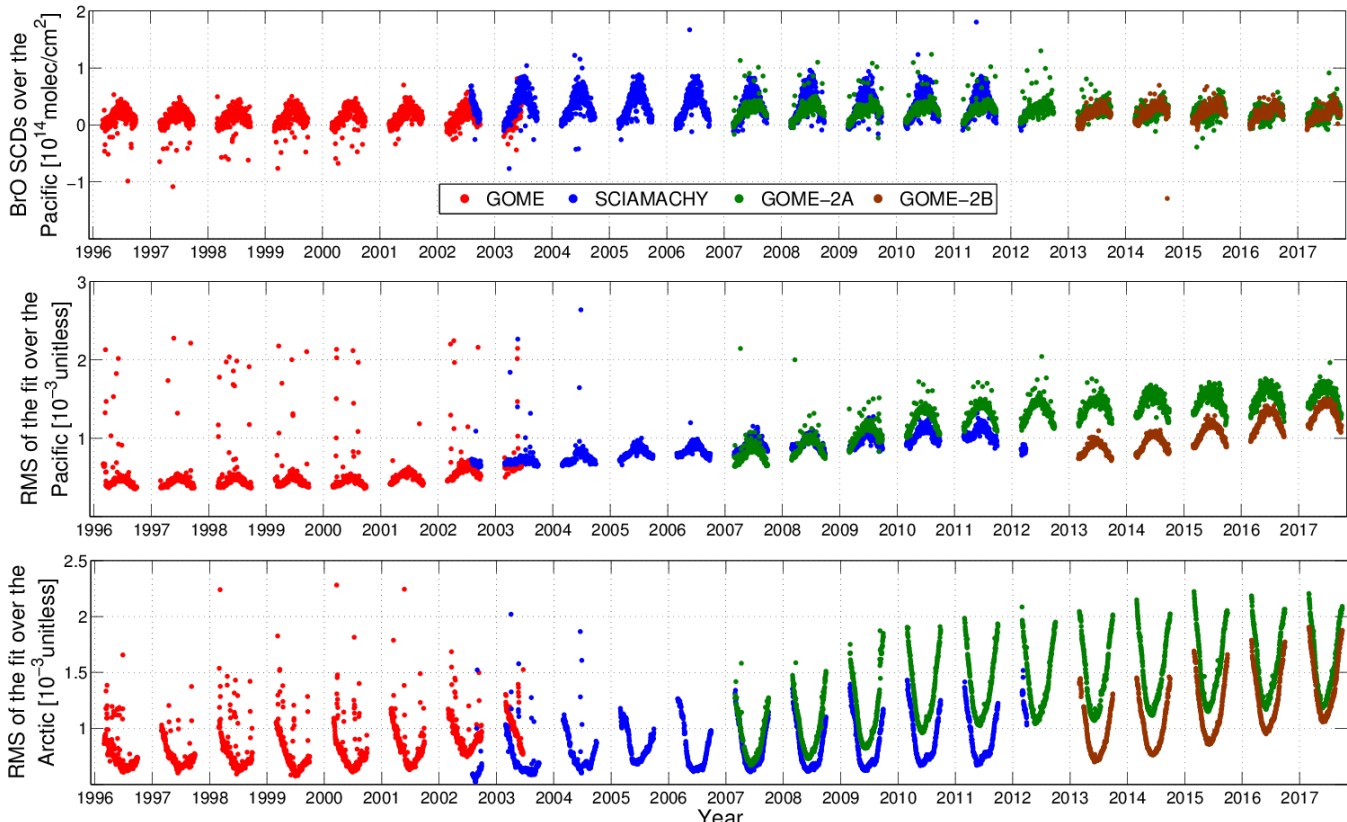

**Figure 1: Time-series of a) SCDs of BrO [molec/cm²] over the Pacific, b) fitting RMS over the Pacific and c) fitting RMS over the Arctic. GOME data is shown in red, SCIAMACHY in blue, GOME-2A in green and GOME-2B in brown colour.**

The annual cycle in the fitting RMS for the Arctic results from changes in the sun's position and the impact on the upwelling
radiance. In spring and autumn the solar zenith angle is larger, the scattering and attenuation increases, and as a result, the
radiance signal is low. In the Arctic, SCIAMACHY has the lowest fitting RMS of all instruments. GOME-2A shows a rapid
increase in RMS until 2009, and a smaller upward trend in the following years. A similar systematic increase in RMS occurs
for GOME-2B. The RMS for the Pacific region shows a similar behaviour for GOME-2A and GOME-2B, as both increase
strongly with time. GOME appears to have lower RMS values on average than the GOME-2A and GOME-2B instruments,
presumably because of the lower spatial resolution, but the RMS shows large variability. Generally, daily mean RMS values
are below $2.0 \times 10^{-3}$, for all instruments. There is no clear trend in the SCD of BrO over the Pacific region for any of the
instruments.

The use of a reference area over the Pacific as background spectrum is an alternative to the use of solar irradiance
measurements, which removes systematic errors arising from interfering instrumental structures in the solar irradiance
measurements. The region of the Pacific is selected because of its relatively small annual cycle of BrO (Richter et al., 2002).
In this way, the quality of the fit improves greatly as problems in the radiances mainly cancel out. This also has other



benefits. For example, GOME-2A is currently drifting in orbit and there are periods during which solar measurements can no longer be carried out. Consequently, the Pacific background spectrum has been used instead of the Sun background spectrum for all instruments in the present study. Although trends over the clean Pacific background region resulting from instrumental degradation were minimised, residual trends had to be accounted for. This has been achieved by applying an

5 additional Pacific correction to each instrument separately. The normalization method computes the average BrO SCD in a small Pacific area ($0.0^{o} \pm 10.0^{o}$ latitude, $180.0^{o} \pm 20.0^{o}$ longitude) and then subtracts this average from every pixel of the BrO SCD. To compensate the negative bias imposed by the method, a constant offset of $7 \times 10^{13}$ molec/cm$^{2}$ was added for every day (Richter et al, 2002; Sihler et al., 2012). This correction tackles offset errors that are occurring for weak absorbers, and are due to instrumental degradation (Alvarado et al., 2014).

10 The empirically determined set of parameters used in the fitting of BrO in the spectral windows are reported in Tables 2 and 3.

**Table 2: Parameter selection for all instruments**

| Parameters | Cross sections - Application Selection |
|---|---|
| **Ozone** | (Serdyuchenko et al., 2014), 223K and 243K |
| **BrO** | (Fleischmann et al., 2004), 228 K |
| **Ring effect** | Ring cross section calculated by SCIATRAN model |
| **Under sampling correction** | Yes |
| **Fraunhofer atlas** | Chance and Kurucz (Chance et al., 2010) |
| **Background spectrum** | Pacific area (-50.0$^{o}$ to 10.0$^{o}$ lat., 235.0$^{o}$ to 270.0$^{o}$ lon.) |
| **Degree of the polynomial** | 4$^{th}$ |

**Table 3: Fitting windows used for the different instruments**

| Instrument | Fitting Window [nm] |
|---|---|
| **GOME** | 336.8 - 358 |
| **SCIAMACHY** | 336 - 347 |
| **GOME-2A** | 337.5 - 357 |
| **GOME-2B** | 338 - 360 |



### 3.2 Tropospheric BrO vertical column densities and datasets

$NO_2$ and $O_3$ column satellite retrievals, tropopause height from meteorological reanalysis data are used for extracting the tropospheric BrO component from the retrieval (stratospheric separation). Sea ice data (age and type) obtained by satellite remote sensing was used in order to identify regions with sea ice cover and hence high surface reflectivity required for the retrieval of tropospheric BrO in this study and for data interpretation in relation to bromine sources.

### 3.2.1 Stratospheric Separation

In order to derive the tropospheric BrO VCD from the retrieved SCD of BrO, the method of Theys et al. (2009) was used. Briefly, this method is based on a stratospheric BrO climatology derived with the BASCOE model (Errera and Fonteyn, 2001) and requires year, latitude, tropopause height, $O_3$ and $NO_2$ columns as input. This approach has been applied successfully in previous studies (e.g. Begoin et al., 2010; Theys et al., 2011; Blechschmidt et al., 2016; Choi et al., 2018). In the present study, satellite derived $O_3$ VCDs from Weber et al. (2013), stratospheric $NO_2$ VCDs from the QA4ECV project (Boersma et al., 2017), and from the Tropospheric Emission Monitoring Internet Service (TEMIS), (Boersma et al., 2004) and tropopause heights derived from NCEP reanalysis (Kalnay et al., 1996) data were used as input. In Theys et al. (2009), a correction factor was applied, to account for the long-term reduction of bromine emissions in the stratosphere. This factor was based on ground based zenith-sky measurements of BrO over Harestua (Hendrick et al., 2008). In the present study, this factor was excluded, as the long-term development of stratospheric BrO VCDs of the model without applying the correction factor comes to a closer qualitative agreement with updated measurements of BrO over Harestua (from F. Hendrick, BIRA-IASB, personal communication). The time-series of $NO_2$, $O_3$ and tropopause height are shown in Figure 2:



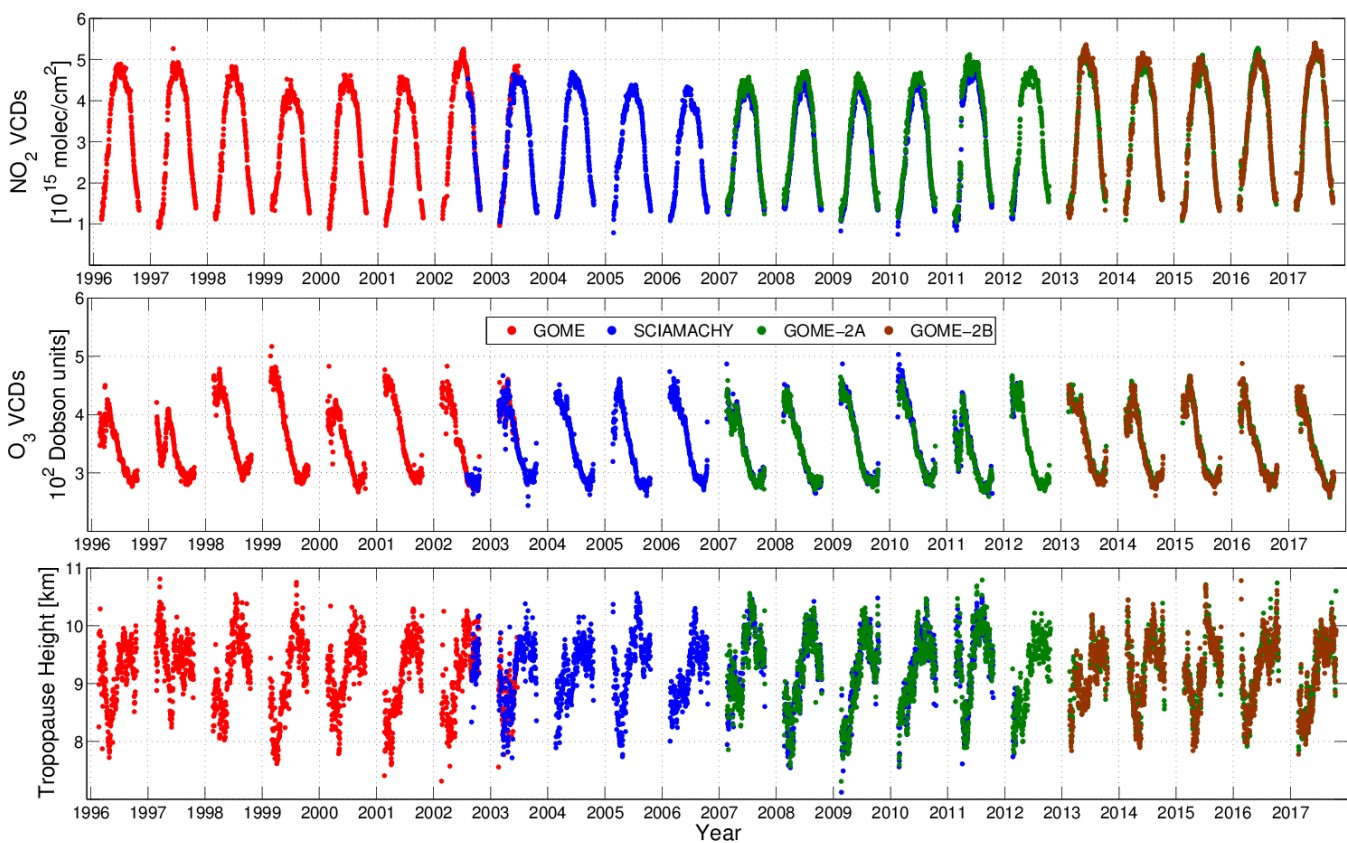

**Figure 2: Time-series of daily averaged input data over the Arctic used for deriving stratospheric BrO VCDs: a) stratospheric NO₂ VCDs [molecules/cm²] from QA4ECV (GOME, SCIAMACHY and GOME-2A) and TEMIS (GOME-2B), b) O₃ VCDs [DU] from Weber et al. (2013) and c) tropopause height [km] from NCEP reanalysis data. Data for the GOME instrument is coloured in red, SCIAMACHY in blue, GOME-2A in green and GOME-2B in brown. All three time-series show daily averages over the Arctic region (>70°N).**

The following formula (Theys et al., 2009) is used to derive the tropospheric VCD of BrO:

$$VCD_{tropo} = (SCD_{total} - VCD_{strato} \times AMF_{strato}) / AMF_{tropo} \qquad \text{Eq. (6)},$$

where $SCD_{total}$ is the slant column of BrO retrieved by the DOAS method, $VCD_{strato}$ corresponds to the stratospheric BrO VCD derived from the Theys et al., (2009) climatology, $AMF_{strato}$ is a stratospheric air mass factor, $AMF_{tropo}$ is a tropospheric air mass factor. For the latter, as in Begoin et al. (2010) and Blechschmidt et al. (2016), a surface albedo of 0.9 has been assumed above sea ice and that all tropospheric BrO is well mixed within the boundary layer extending to 400m altitude. Note that the stratospheric BrO VCD column is independent of the BrO SCD derived from the DOAS retrieval described in section 3.1 and settings therein and only depends on the Theys et al. (2009) climatology and its inputs.



### 3.2.2 Sea Ice

In order to study the connection between tropospheric BrO and sea ice under the impact of Arctic Amplification, long term sea ice data (starting from 1996) is required in this study. In addition, since the tropospheric AMF applied for the retrieval of tropospheric BrO (see previous section) assumes a surface albedo of 0.9, the sea ice data was used to remove data with no sea ice cover from the BrO data. For this purpose, the sea ice age dataset from Tschudi et al. (2019) was used. It is retrieved from different passive microwave satellite remote sensing instruments and has a very high spatial resolution of 12.5x12.5 $km^2$, while its temporal resolution is 7 days.

### 4 Results

In this study, the focus is the long term analysis of BrO columns during the period from March until September. This period has the best coverage by the UV-VIS satellite remote passive sensors at high latitudes. The months of October to February have few or no observations due to polar night in the target latitudes. The analysis is restricted to data northwards of $70.0^0$ referred to as the Arctic here. This region was chosen because the source regions for BrO explosion events are known to be associated with regions of high sea ice cover. We present the derived BrO time-series for the Arctic region, together with corresponding annual cycles, scatter plots and map plots in section 4.1. The tropospheric BrO is compared to sea ice coverage and age in section 4.2, to investigate the impact of changing sea ice conditions due to Arctic Amplification on BrO amounts in the Arctic's troposphere. Finally, a trend analysis of tropospheric BrO is performed, together with a check for statistical significance of the trends (section 4.3).

### 4.1 Comparison between the BrO columns derived from different sensors

The geometric BrO VCD is derived by dividing the BrO SCD with a simple stratospheric AMF, which takes into account the scattering at the surface but ignores the impact of scattering within the atmosphere. Consequently, it necessarily differs from the sum of the tropospheric and stratospheric column. Daily values of the geometric, stratospheric and tropospheric VCDs of BrO over the Arctic region are shown in Figure 3.

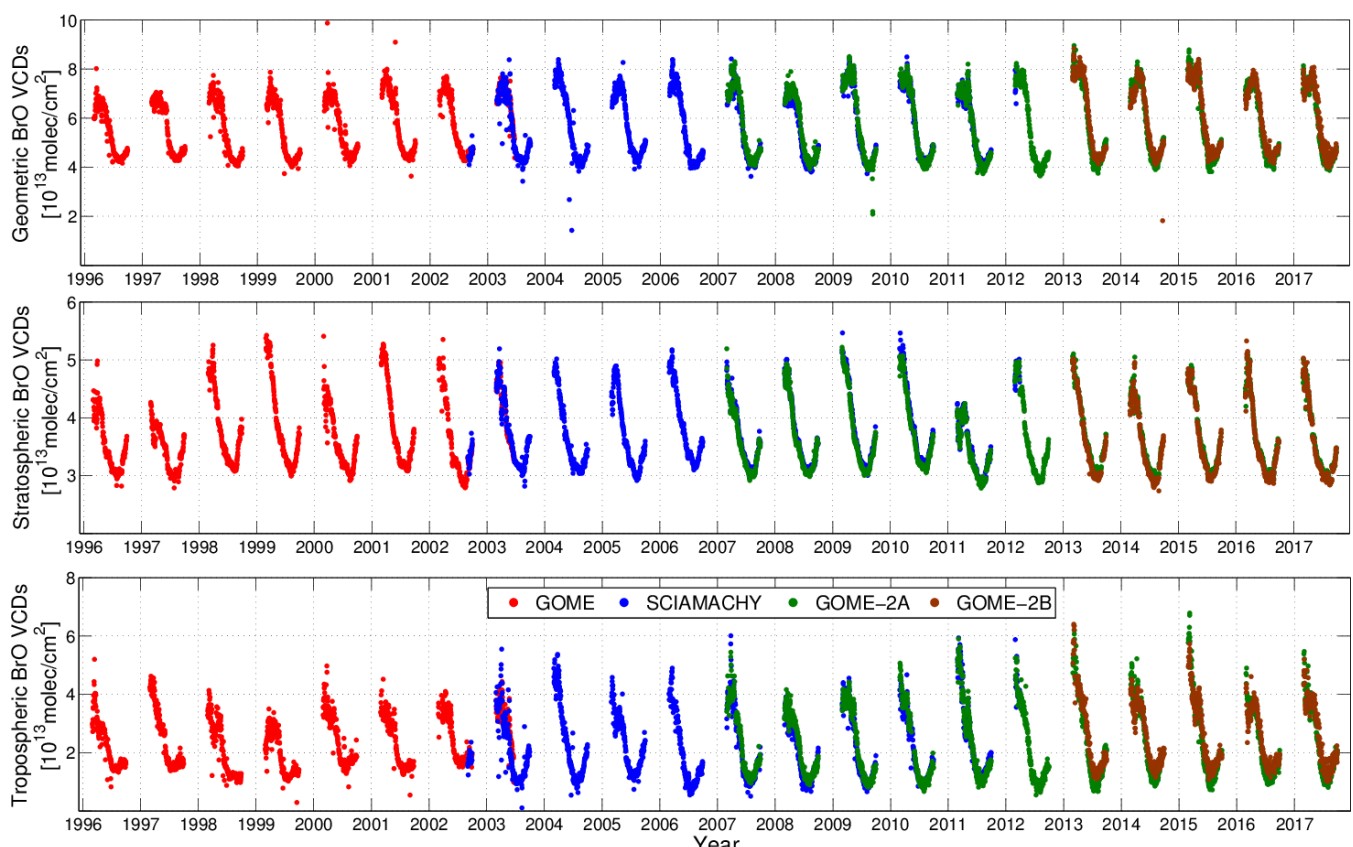

**Figure 3: Long-term BrO time-series over the Arctic region: a) daily geometric BrO VCDs [$10^{13}$ molec/cm$^2$], b) daily stratospheric BrO VCDs [$10^{13}$ molec/cm$^2$] and c) daily tropospheric BrO VCDs [$10^{13}$ molec/cm$^2$]. All figures show daily averages $\geq 70.0^\circ$ N latitude. GOME data is coloured in red, SCIAMACHY in blue, GOME-2A in green and GOME-2B in brown.**

5  The geometric VCD accounts for the differences from the different viewing geometries of the satellite sensors to a first order. Qualitatively, there is good agreement during periods of overlapping measurements e.g. between GOME and SCIAMACHY (from August 2002 to June 2003), SCIAMACHY and GOME-2A (from March 2007 to March 2012) and finally GOME-2A and GOME-2B (from March 2013 to September 2017). A quantitative comparison is provided below. The stratospheric BrO VCDs show a small upward trend from 1995 to 2001 and a slight decrease afterwards as a result of

10  changes in $O_3$, $NO_2$ and tropopause height.

The comparison of tropospheric BrO VCDs time-series during the periods of overlap between the different instruments show a similar level of agreement to that for the geometric VCDs. The seasonality of the tropospheric time-series is also similar to the geometric one. We attribute this to the inorganic release of bromine associated with sources, which depend on sea ice and meteorological parameters. As described in the introduction, in polar spring, the combination of low temperatures on first

15  year sea ice, presumably having sufficient brine, triggers the release of Br in the atmosphere. This occurs dominantly in spring. During polar summers and early autumn, the sea ice extent reaches the yearly minimum and the temperature its





maximum, because of the increased solar insolation. The release of bromine compounds from the ocean is hence expected to be highest in summer and early autumn. This provides a biogenic source of bromine, but the oxidation of organic bromine containing compounds is relatively slow compared to the release of BrO from BrO explosion events. During summer and early autumn, tropospheric BrO VCDs reach their minimum values of the year. A small increase is observed in each

September. The origin of this increase is not yet identified. Potential candidate explanations include inaccuracies in the stratospheric BrO VCDs calculation or potentially lower temperature accelerating the inorganic release of bromine from brine without reaching the threshold for explosion.

We identify peaks in the tropospheric BrO VCDs time-series: 2007, 2013 and 2015 are the years with the highest tropospheric BrO VCDs in polar spring. Hollwedel et al. (2003) investigated the period from 1996 to 2001. Despite the

different settings and cross sections used in the retrieval, their geometric VCDs have similar magnitudes and patterns to those presented here (i.e. they increase from 1997 to 1998 and then decrease from 1999 to 2000). It is also interesting to compare the results from this study with those from Choi et al. (2018). They used the operational product of the OMI instrument (from 2005 to 2015) and the same stratospheric separation method as applied here. In agreement with our findings, Choi et al. (2018) found a peak in tropospheric BrO VCDs in 2007. However, the OMI dataset does not show the

same increase as the SCIAMACHY and GOME-2A data in later years. The origin of this difference is not identified. It may be due in part to the strong row anomaly affecting the OMI data product (Class et al., 2010).

Figure 4 shows scatter plots of geometric and tropospheric VCDs, for the three overlapping time periods of the instruments.



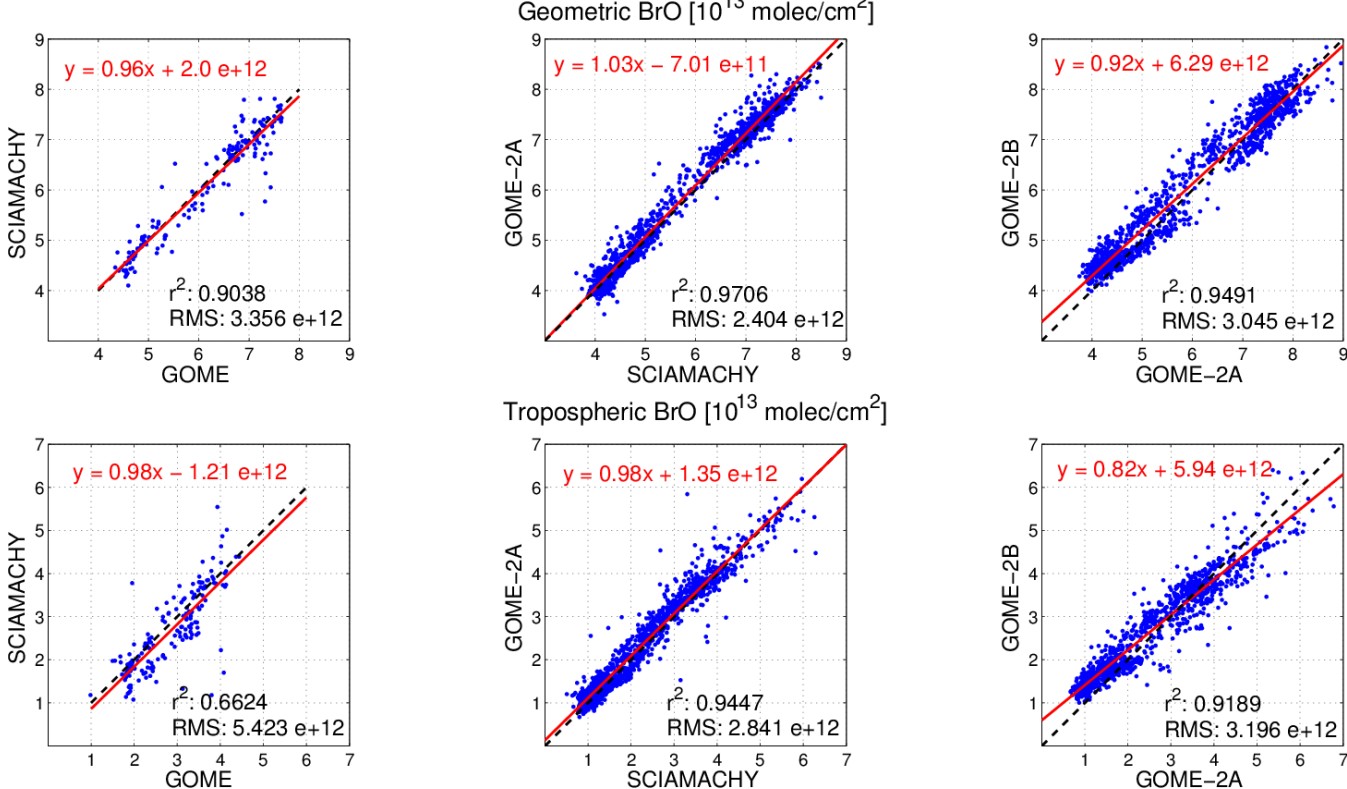

**Figure 4: Scatter plots of geometric BrO VCDs (upper row), and tropospheric BrO VCDs (lower row) for (left) GOME against SCIAMACHY (August 2002 to June 2003), (middle) SCIAMACHY against GOME-2A (March 2007 to March 2012) and (right) GOME-2A against GOME-2B (March 2013 to September 2017). The dashed black line in each scatter plot is the reference line, the red one is the linear regression line. The Pearson correlation coefficient (squared, $r^2$), the RMS and the y function of the regression line are also given. The units for all scatter plots are $[10^{13}$ molecules/cm$^2]$.**

The best agreement is found between SCIAMACHY and GOME-2A, both for the geometric and the tropospheric VCDs, while the least good agreement is found between GOME and SCIAMACHY (this can be attributed to the comparably short overlapping period between these sensors).

10 Climatological seasonal cycles (averages over the whole period of retrievals of geometric, stratospheric and tropospheric VCDs) for each instrument are shown in Figure 5.





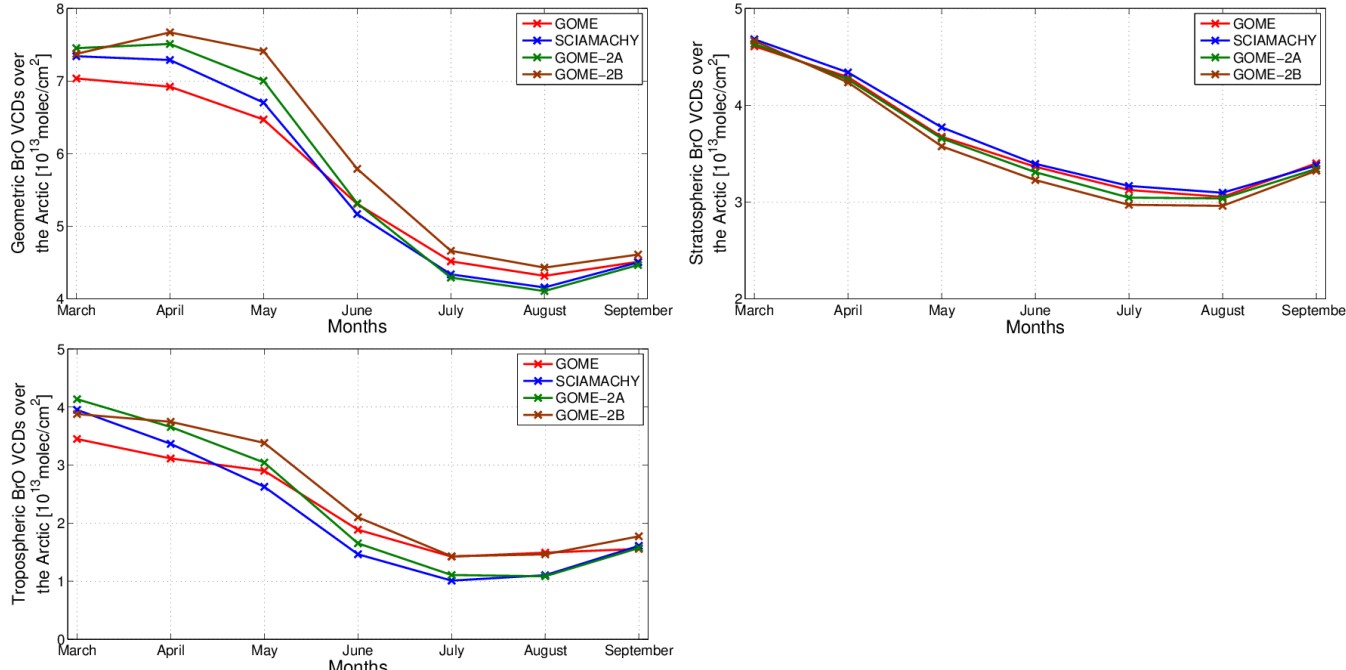

**Figure 5: Climatological seasonal cycles of a) geometric BrO VCDs [$10^{13}$ molec/cm$^2$], b) Stratospheric BrO VCDs [$10^{13}$ molec/cm$^2$], and c) Tropospheric BrO VCDs [$10^{13}$ molec/cm$^2$] over the Arctic for GOME (red), SCIAMACHY (blue), GOME-2A (green) and GOME-2B (brown).**

5    The annual cycles of the geometric and the tropospheric VCDs time-series are similar in shape. The largest VCDs occur for all instruments in polar spring (March to May), followed by a strong decrease from May to June. In spring, GOME-2B VCDs are slightly larger, while GOME columns are slightly lower than data from the other instruments for the geometric VCDs. In summer and early autumn GOME VCDs are higher. For the tropospheric VCDs, the differences between the instruments become smaller.

10   Figure 6 shows average maps of geometric VCDs of BrO over the Arctic for the overlapping periods of the sensors.

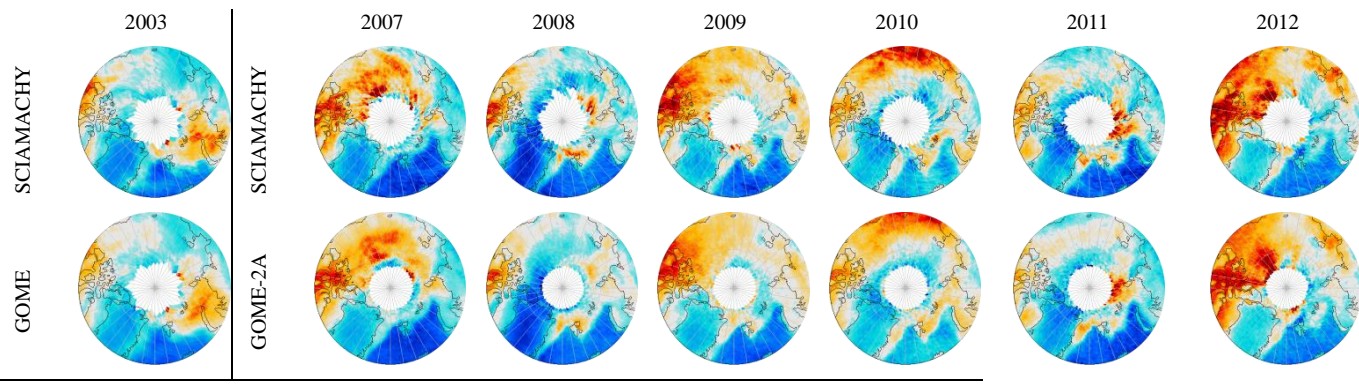



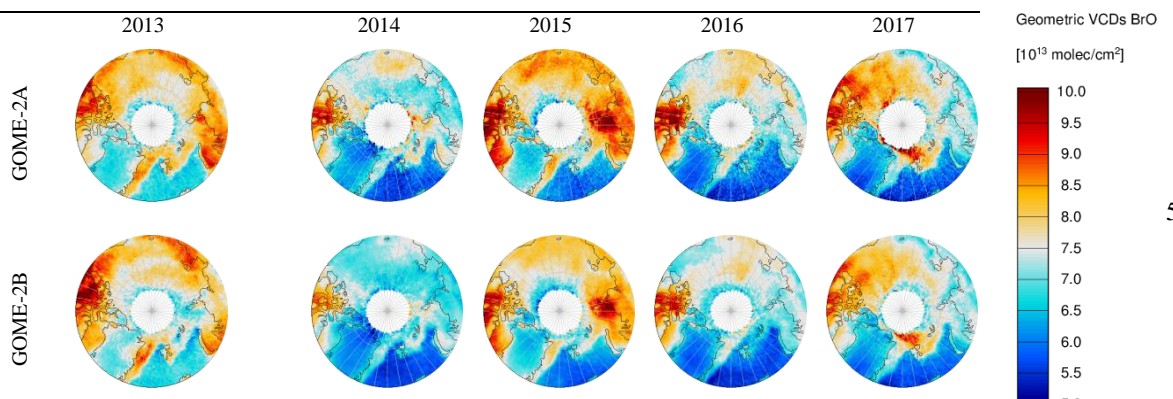

**Figure 6: Monthly mean for March of geometric BrO VCDs [$10^{13}$ molec/cm$^2$], in the Arctic region. Rows indicate different instruments, while columns different years, corresponding to overlapping periods of instruments.**

We see that for most years, high geometric BrO VCDs occur in similar areas (and are of similar magnitude) for the different

10  instruments. A qualitative comparison of geometric BrO VCDs between this study and the one by Hollwedel et al. (2003) has been performed, showing again good agreement, with plumes of BrO being of similar magnitude and appearing over the same regions.

Figure 7 shows selected tropospheric BrO VCD maps over the Arctic for particular BrO explosion events, one for each overlap period of the sensors. For every event, we present 4 days, where BrO was released and transported in the Arctic

15  troposphere.





**Figure 7: Examples of BrO explosion events over the Arctic region for overlapping periods between the sensors. Rows refer to the different instruments, columns refer to the dates. For each case, four daily average maps of tropospheric BrO [$10^{13}$ molec/cm$^2$] are shown.**

BrO explosion events are characterised by high values of tropospheric BrO VCDs, which occur every polar spring. The agreement between the tropospheric BrO VCDs retrieved from the different sensors is good, as shown above. BrO plumes appear over the same areas and have similar magnitudes for each bromine explosion case. The times of observations are similar but not identical and as yet, we have not identified evidence of this impacting the comparison of tropospheric BrO VCDs. The transport of plumes of BrO by cyclones away from the initial release area (e.g. the 2007 case shown in Figure 5, investigated in more detail by Begoin et al., 2010) is also observed by the sensors. Hence, the track of cyclones transporting BrO plumes is expected to influence longer-term averaged BrO maps.



## 4.2 Arctic Tropospheric BrO and its Relation to Sea Ice

In this section, the evolution of tropospheric BrO and its relation to sea ice is presented. More specifically, the relationship between first year ice and tropospheric BrO is investigated. Since an albedo of 0.9 was assumed in the tropospheric AMF, a sea ice based ground scene flagging of our tropospheric BrO dataset was performed. In this way, only BrO observations

5 above sea ice are analysed. As described in section 4.1, the individual BrO VCDs time-series from the four sensors are highly consistent and the remaining inconsistencies are small. Hence, a merged tropospheric BrO dataset over sea ice was derived, by simply averaging the overlapping days between the sensors. The merged tropospheric VCDs BrO dataset is shown in Figure 8 together with the results from the individual sensors and the time-series of sea ice extent from Tschudi et al. (2019).

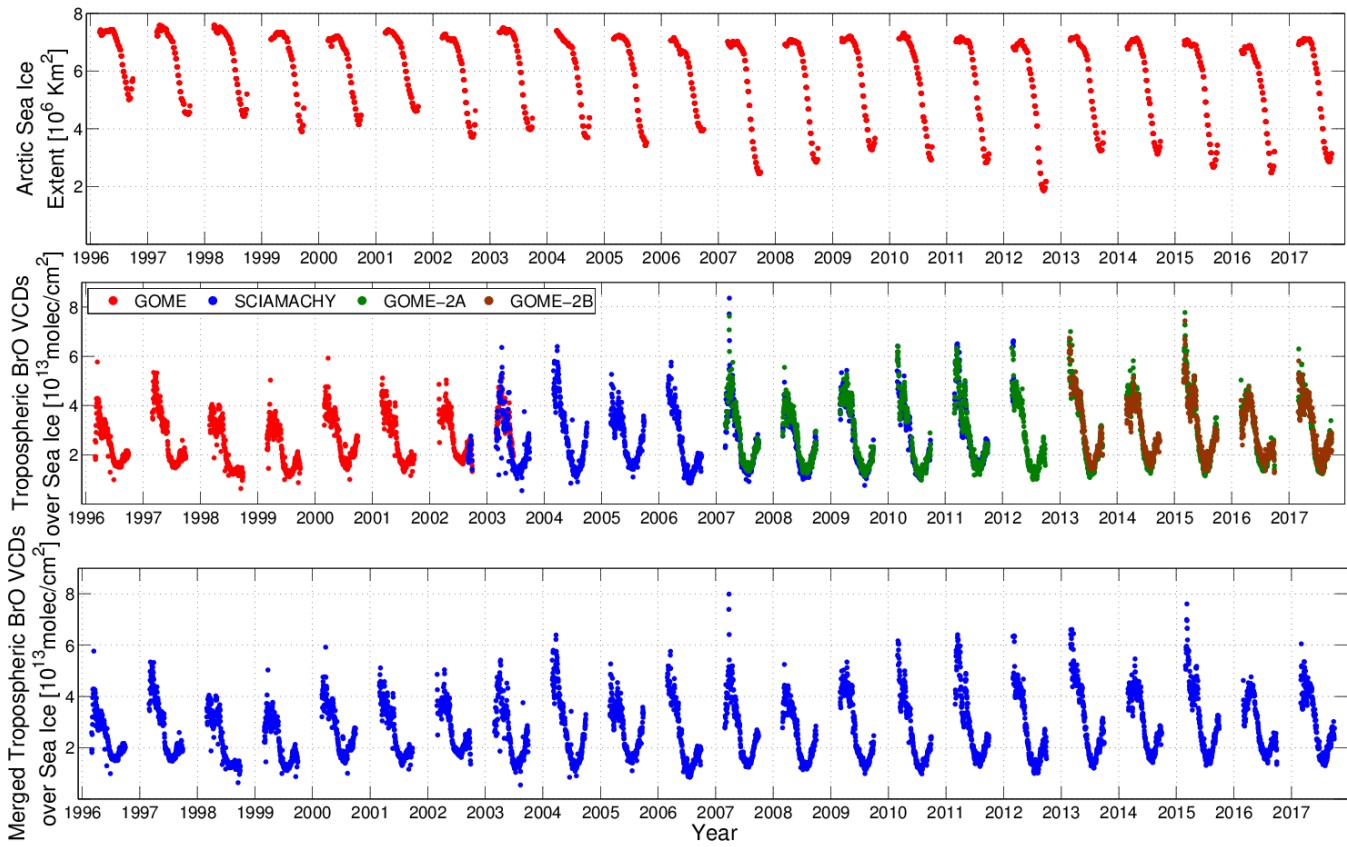

**Figure 8: a) Arctic sea ice extent, from March till September from Tschudi et al. (2019), b) tropospheric BrO VCDs above sea ice in the Arctic. (GOME data is shown in red, SCIAMACHY in blue, GOME-2A in green and GOME-2B in brown), c) the sensor merged tropospheric BrO VCDs above sea ice in the Arctic.**

By comparing Figures 8b and 3c, it is found that BrO explosion events are becoming more evident (for example in years

15 2007 and 2015) if we apply the sea ice flagging (i.e. keeping only scenes with sea ice coverage).





In order to investigate the spatial evolution of tropospheric BrO in time and its relationship to sea ice age, polar spring (March to May) average maps of tropospheric BrO and corresponding sea ice age maps are displayed in Figure 9 for each year between 1996 and 2017.





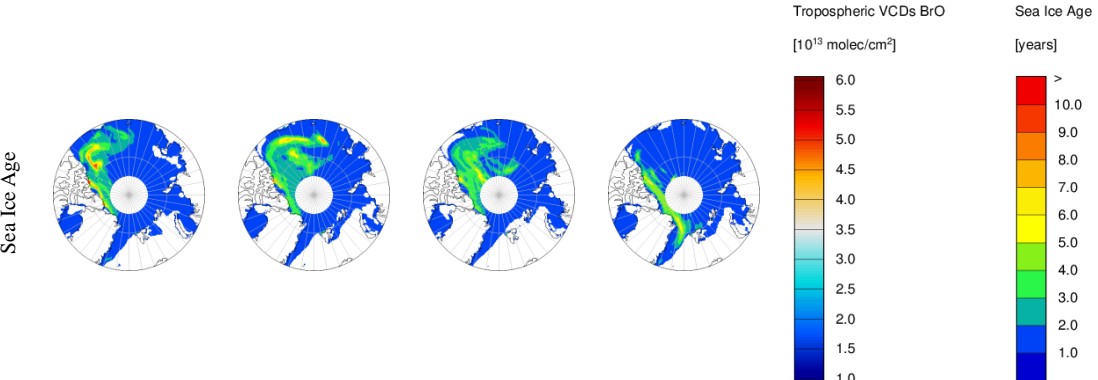

**Figure 9: Polar Spring (MAM) averages of tropospheric BrO VCDs [10$^{13}$ molec/cm$^2$] over sea ice, compared to sea ice age in the Arctic. Columns refer to years, odd rows to tropospheric BrO VCDs and even rows to sea ice age.**

We find that BrO plumes have intensified over the latest years (especially during 2013-2015). In addition, the regions in which BrO explosion events occur have changed over the years. For example, over the last years (i.e. 2010 and onwards), BrO is formed at the eastern coastline of Greenland and inside the Arctic Ocean (latitudes larger than 80.0°), something not evident in the early years (i.e. 1996 to 2001). Also, in the early years (i.e. 2001 - 2003), most of the BrO is found in the

5 region of the Barents and Kara Sea (to the north of Scandinavia and Russia). However, over the last years (2009 and onwards), BrO plumes "have spread" over the Arctic region, even appearing at comparatively lower latitudes, to the eastside of Greenland.

First year ice has become more dominant over the Arctic region in recent years (i.e. 2009 onwards). 2009 was the first year for which large tropospheric BrO VCDs were evident over the Canadian archipelago and the Beaufort Sea. In 2009, the area

of multi-year ice decreased significantly in these regions. Similarly, the appearance of BrO plumes over the north-west coast of Greenland in 2015 is in agreement with the appearance of first year ice during this year. The relationship between first year ice and BrO plumes is not straightforward or linear and apparent in all years. This can be explained by transport of BrO and of BrO sources (such as blowing snow and cold brine). In 2004 for example, the highest BrO VCDs are found over a multi-year ice area, while in 2002 BrO is mainly seen over first year ice. The increase of magnitude of BrO plumes is not

simply related to the development of first year ice as identified in the maps above. In summary, the increase of the areas where tropospheric BrO explosions occurs is related in a complex way to the increase of first year ice covered Arctic regions.

Figure 10 shows yearly anomaly polar spring maps of tropospheric BrO VCDs with respect to the 22 year mean. The 22 year average is shown in the last row of the Figure:





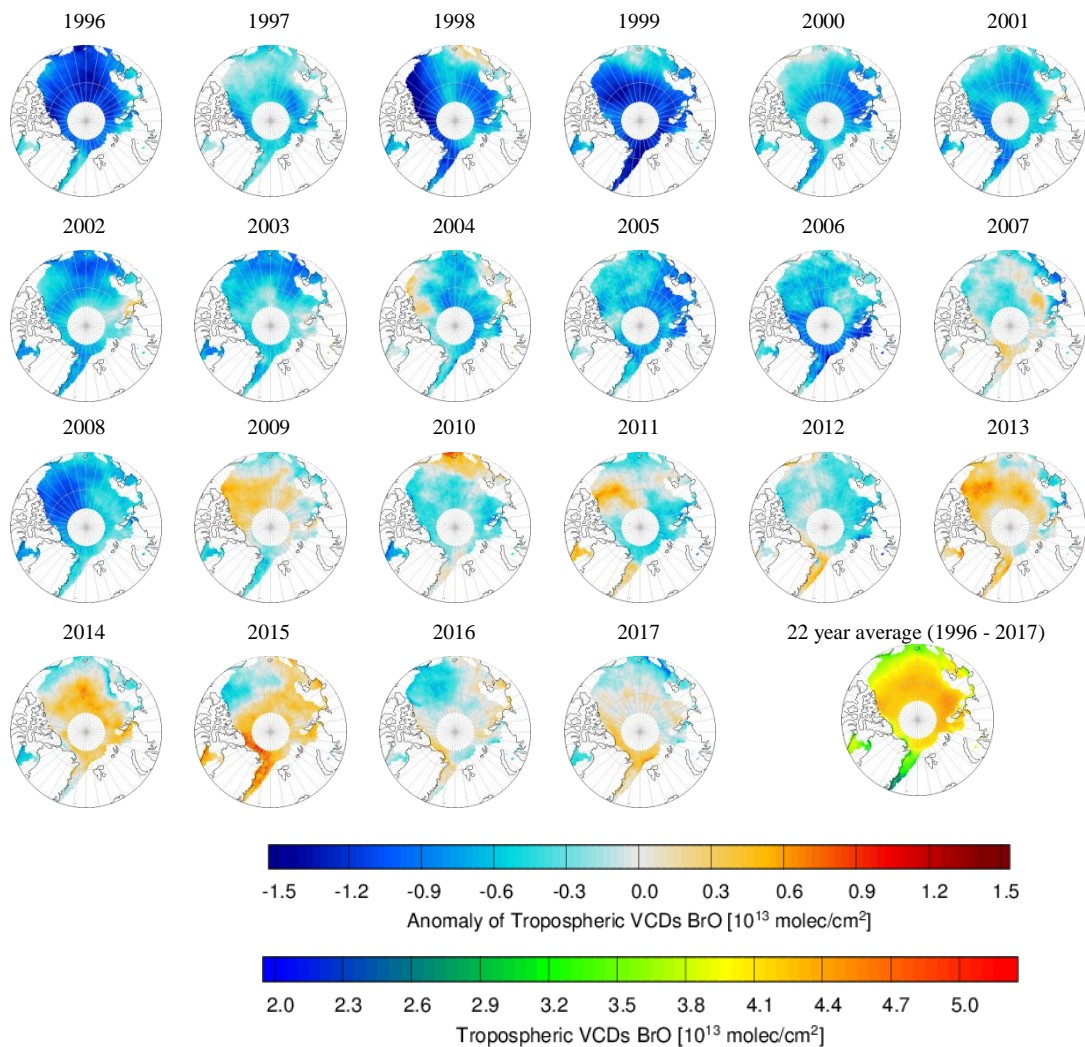

**Figure 10: Anomaly maps of polar spring (MAM) tropospheric BrO [$10^{13}$ molec/cm$^2$] over sea ice. From every MAM average, the 22 year average (1996 to 2017, shown at the bottom right of row four) of tropospheric BrO VCDs was subtracted.**

During the first years of the time-series, BrO VCDs are lower than on average in most regions of the Arctic. The first large positive anomaly compared to the mean occurs in 2009, and in areas where first year ice appeared for the first time.

Over the last years (2011 onwards), positive differences appear to the north and to the east of Greenland. Overall, and when comparing the last with the first five years, we see that the increase of BrO plumes is apparent over the last years of Arctic

5    Amplification.

Figure 11 shows plots which probe the relationship between tropospheric BrO and first year ice extent. More specifically, Figure 11a shows polar spring tropospheric BrO over sea ice and first year ice extent, Figure11b shows the area extent where tropospheric BrO exceeds the threshold of $7 \times 10^{13}$ molecules per cm$^2$. This threshold value was chosen empirically, as in previous studies (Hollwedel et al., 2003; Choi et al., 2018). First year ice extent, and corresponding scatter plots are also





given. For the time-series of this figure, all the polar spring values for each year were averaged, thereby deriving one value

per year.

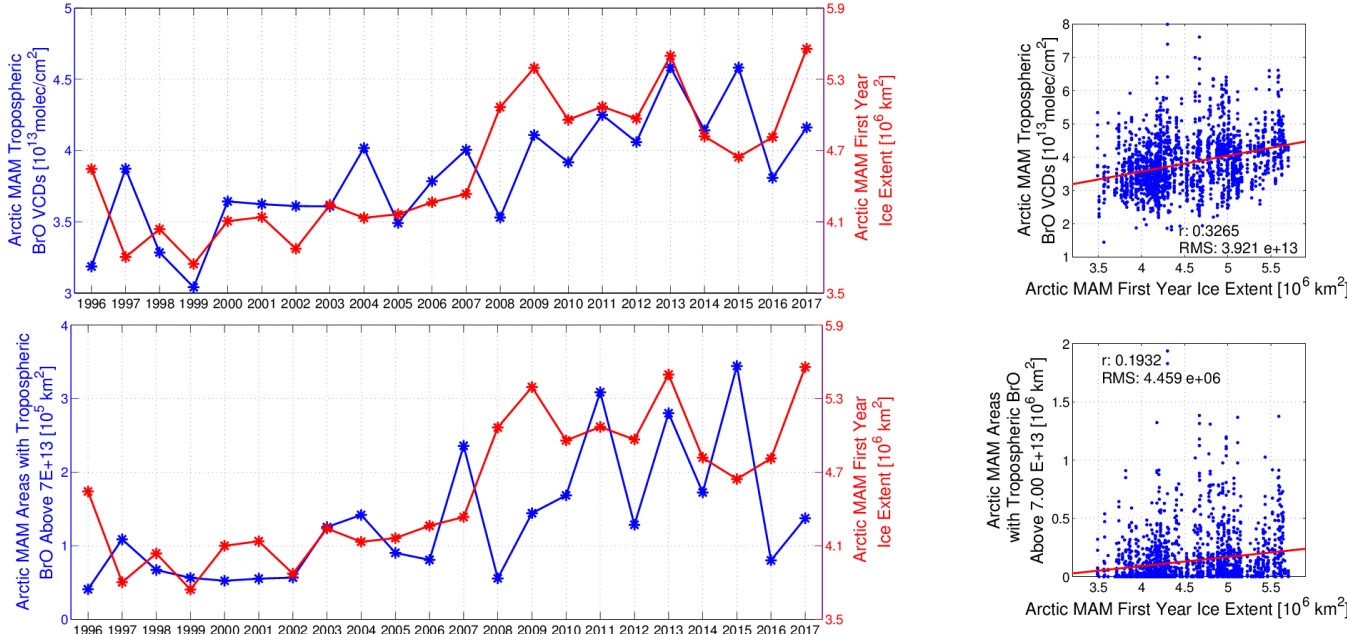

**Figure 11: a) Polar spring (MAM) mean time series of tropospheric BrO VCDs over sea ice and first year sea ice extent over the
Arctic, b) Polar spring (MAM) mean time series of areas with BrO VCDs above the threshold of $7x10^{13}$ molec/cm$^2$ and first year
sea ice extent over the Arctic, c) scatter plots showing polar spring averages of tropospheric BrO VCDs against first year ice extent
and d) polar spring averages of areas with tropospheric BrO VCDs exceeding the threshold of $7x10^{13}$ molec/cm$^2$ against first year
ice extent in the Arctic. The linear regression line is shown in red in each scatter plot.**

The total Arctic sea ice extent has decreased over the last years, but the area covered by first year sea ice has increased. The

type of ice encountered is hence changing. Due to the increasing temperatures in the Arctic, more sea ice melts every year,

and more fresh or young sea ice is formed each winter period. Figure 11a shows that 2008 was the first year that first year

ice extent exceeded the threshold of $5x10^6$ km$^2$, which is also the case for almost all of the following years. During the

decade from 2007 to 2017, first year ice extent was larger during polar springs than from 1996 to 2006. Although

tropospheric BrO VCDs seem to have also increased in magnitude approximately from 2007 onwards, the yearly evolution

between the two does not correlate strongly. For example, for years 2008 and 2016 a decrease of tropospheric BrO VCDs

compared to the previous year is found, while first year ice extent increased. In 2015, the highest tropospheric BrO VCDs

polar spring averages are found. In the same year, the polar spring first year sea ice extent average has its lowest value after

2008. Similar conclusions can be drawn from the top right scatter plot (Figure 11c): Although most days of average BrO

above $5x10^{13}$ molec/cm$^2$ are occurring with first year ice extent above $4.5x10^6$ km$^2$, we can see such days also below this

threshold. The correlation coefficient between polar spring tropospheric BrO VCDs and first year sea ice extent for the same

period is 0.32. The correlation between areas of enhanced tropospheric BrO VCDs and first year ice extent is even lower,





approximately 0.2. This is evident from both the time-series (Figure 11b) and their scatter plot (Figure 11d, bottom right). It seems that in both cases, the largest number of days with high tropospheric BrO VCDs or largest areas of BrO explosion events are occurring when first year ice extent is between 4.5 and $5 \times 10^6$ km$^2$, and not when it reaches its peak. The magnitude of tropospheric BrO VDs is not closely related to first year ice extent. This agrees with the findings of Choi et al, (2018), where a similar analysis is provided based on OMI data. They found a correlation coefficient of -0.32 between first year ice extent and tropospheric BrO explosion frequency. The negative sign may be attributed to the decrease of their tropospheric BrO VCDs over the latest years, which arguably is linked to degradation of the OMI instrument.

### 4.3 Trend analysis

Finally, a trend analysis of the tropospheric BrO VCDs time-series is performed, in order to investigate the statistical significance of the observed temporal and spatial changes in tropospheric BrO VCDs described above in relation to Arctic Amplification. The tropospheric BrO VCDs time-series shows a seasonality, with a maximum every polar spring (March in most cases) and a minimum every summer. Consequently, a model which combines a linear trend and seasonal components is selected, as in other studies (Hendrick et al., 2008; Georgoulias et al., 2018), to calculate the trends appearing in our time-series, expressed by the following formula:

$$d(t) = At + B + \sum_{i=1}^{3} \left\{ C_i \cos\left(\frac{2\pi}{M}(t)\right) + D_i \sin\left(\frac{2\pi}{M}(t)\right) \right\}$$     Eq. (7)

where A is the slope, B the intercept, d(t) is the modelled value of a given day, t is the day of the dataset (expressed in fractional years) and M is the time period. It should also be stated, that the number of harmonic functions was chosen based on the minimization of the residuals between the model and the dataset. Furthermore, the error of each trend was calculated, based on the formula:

$$\sigma_B = \left[ \frac{\sigma_M}{M^{3/2}} \sqrt{\frac{1+\varphi}{1-\varphi}} \right]$$     Eq. (8)

where, $\sigma_M$ is the standard deviation of the residuals between the model and the time-series, M is the period in years, and $\varphi$ is the autocorrelation of the residuals. Finally, a trend is considered significant if the ratio between it and its error is greater than 2 (Weatherhead et al., 1998).

Table 4 shows trends of tropospheric BrO over sea ice over the whole period (see Figure 8c), for individual months and for BrO explosion extent (Figure 11b).

**Table 4: Trends of tropospheric BrO (over the whole period and for individual months) and for BrO explosion extent between 1996 and 2017, together with their errors and significance of each trend.**

| | Units | Trend [units/year] | Error in trend [units/year] | Trend percentage (%/year) | Significant |
|---|---|---|---|---|---|
| **Merged BrO (Figure 8c)** | [molec/cm$^2$] | + $2.4 \times 10^{11}$ | $1.1 \times 10^{11}$ | + 0.99 | Yes |
| **March (Figure 8c)** | [molec/cm$^2$] | + $5.2 \times 10^{11}$ | $8.5 \times 10^{10}$ | + 1.50 | Yes |





| | | | | | |
|---|---|---|---|---|---|
| **April (Figure 8c)** | [molec/cm$^2$] | + 4.9x10$^{11}$ | 5.0x10$^{10}$ | + 1.60 | Yes |
| **May (Figure 8c)** | [molec/cm$^2$] | + 3.9x10$^{11}$ | 5.0x10$^{10}$ | + 1.30 | Yes |
| **June (Figure 8c)** | [molec/cm$^2$] | + 1.5x10$^{11}$ | 4.3x10$^{10}$ | + 0.74 | Yes |
| **July (Figure 8c)** | [molec/cm$^2$] | + 2.9x10$^{10}$ | 2.2x10$^{10}$ | + 0.18 | No |
| **August (Figure 8c)** | [molec/cm$^2$] | + 4.5x10$^{10}$ | 3.1x10$^{10}$ | + 0.26 | No |
| **September (Figure 8c)** | [molec/cm$^2$] | + 2.8x10$^{11}$ | 4.3x10$^{10}$ | +1.43 | Yes |
| **BrO Explosion extent (Figure 11b)** | [km$^2$] | + 896 | 2280 | + 0.06 | No |

As can be seen from Table 4, almost all trends (with the exception of trends of the summer months July and August and for the BrO explosion extent) are significant. The trend of tropospheric BrO over the Arctic shows approximately a 1% increase per year over the last 22 years. However, during polar spring months the increase is stronger, reaching a value of 1.5% per
5 year, while BrO during polar summer (which can be regarded as background BrO in contrast to polar spring when BrO explosion events occur) shows only insignificant change.

Figure 12 shows the trends appearing in the tropospheric BrO time-series, based on different time periods, i.e. different starting and ending years. In contrast to the values derived in Table 4, simple linear trends were calculated (as time periods over multiples of one year are regarded) from and to the same date for both the starting and the ending years (i.e. end of
10 March).

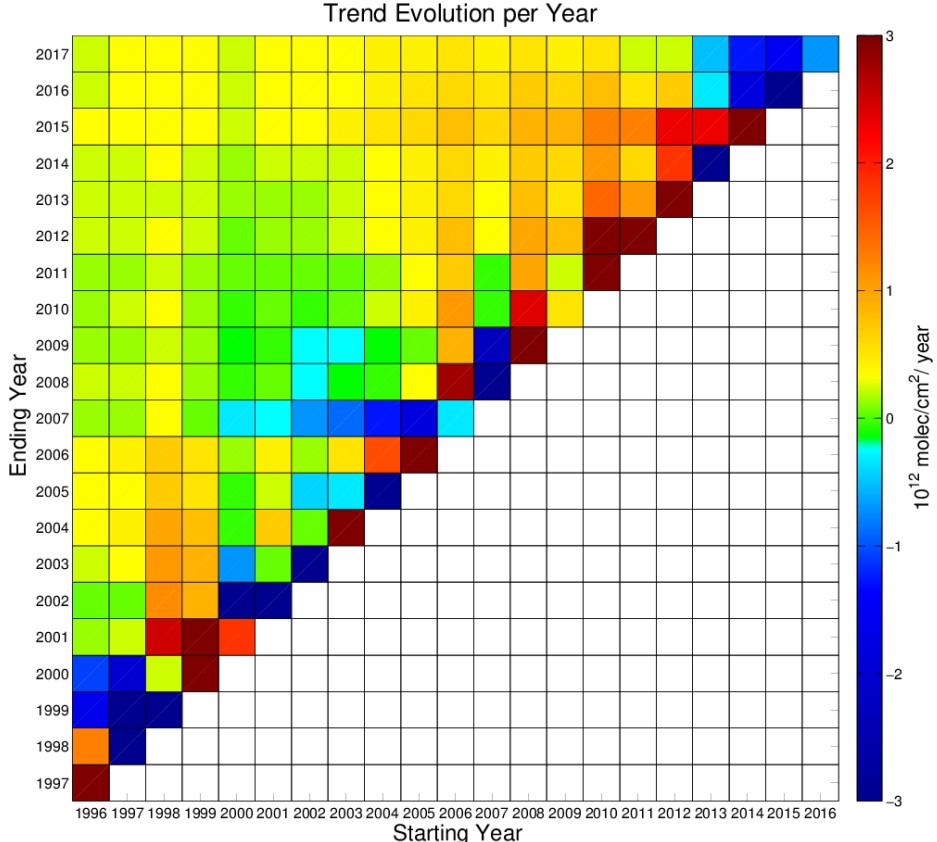

**Figure 12: Trend [10$^{12}$ molecules/cm$^2$/year] evolution of the merged tropospheric BrO dataset (i.e. slope of the linear regression line) over sea ice. The x axis shows the starting year, the y axis shows the ending year of the time period over which the trend was calculated.**

5    Trends starting in one year and ending in the next one (e.g. from 1996 to 1997) are in agreement with Figure 8c (for example, the strong positive trend from 2014 to 2015, or the decrease from 1997 to 1998). Trends over short periods are dominated by inter-annual variability. Mainly positive trends are found, especially for trends covering the latest years as finishing years.

In Figure 13, the trend for each individual grid box during polar spring months is displayed, for both tropospheric BrO and

10   sea ice age. Both maps are gridded in 0.125°x0.125° resolution. The trend for every grid box is calculated based on the method described above, i.e. in Equation 7.





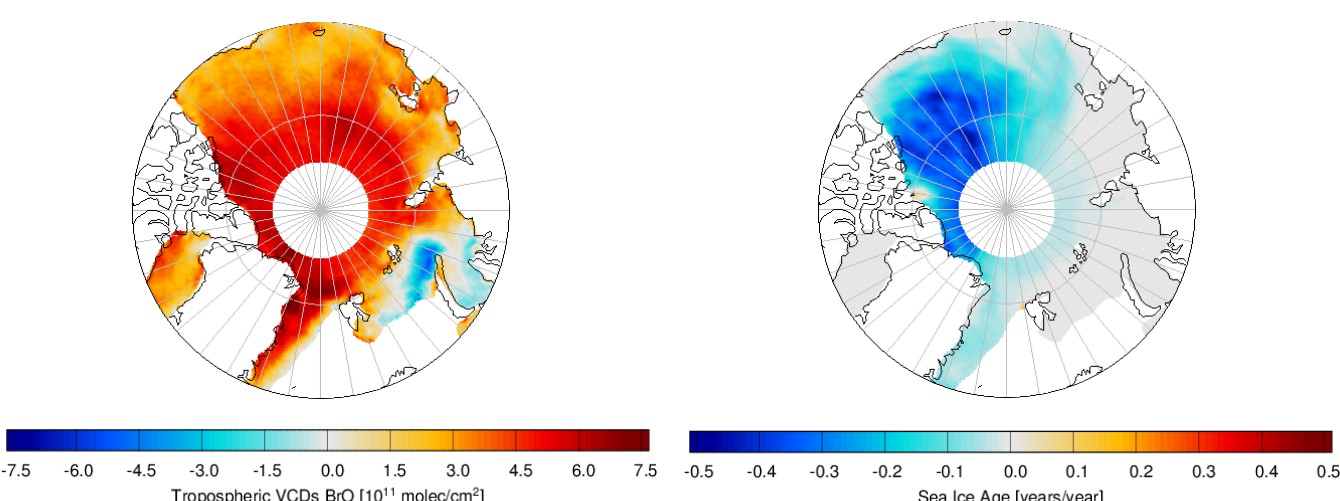

**Figure 13: Maps of the trend for each grid box during polar spring (MAM) for tropospheric BrO over sea ice [$10^{11}$ molecules /cm$^2$ /year] and sea ice age [years /year].**

As shown in Figure 13, tropospheric BrO VCDs values increase inside the region of the Arctic Ocean, and also to the north and east of Greenland, where sea ice age shows a negative trend. From Figure 9, we see that from 2008 and afterwards, first year ice appears in the same area. However, the trends of tropospheric BrO and sea ice age do not match everywhere. The strongest negative trends in the sea ice map occur in areas where multiyear ice was dominant in the past and first year ice is

forming in the latest years (Beaufort Sea and Canadian archipelago). However, in the same area, we do not see the most pronounced increase in the trend map of tropospheric BrO VCDs (the increase is smaller than in other areas). The strongest increase in tropospheric BrO VCDs occurs to the northeast of Greenland close to the Fram Strait, to the north of Greenland and close to Ellesmere Island. However, sea ice age does not show a strong trend to the northeast of Greenland.

## 5 Summary and conclusions

Arctic Amplification, the rapid and intense increase of air temperature over high latitudes is expected to have drastic impacts on all constituents of the Arctic ecosystem. Bromine release from young sea ice, frost flowers, blowing snow and liquid brine was first discovered about 30 years ago. This is of great significance in the Arctic atmosphere, as, through a set of autocatalytic reactions, bromine depletes tropospheric ozone very efficiently, potentially altering the oxidizing capacity of the troposphere. In this study, the first consistent and long-term dataset of tropospheric BrO over the Arctic, retrieved from

four UV-VIS instruments over a 22-year time period is presented. The results demonstrate a high level of consistency of the retrieved BrO VCDs between the different sensors.

Time-series of tropospheric BrO over the Arctic and corresponding trends indicate that the magnitude of tropospheric BrO explosion events has increased over the last years. This increase is pronounced in polar springs, when BrO explosion events





occur. Additionally, by studying the spatial patterns of tropospheric BrO, it can be seen that the size of the areas where BrO explosions occur has increased.

In order to investigate the mechanisms behind these trends, tropospheric BrO and sea ice age time-series and spatial trends were evaluated. The overall increase in tropospheric BrO is, in theory, in agreement with the increase in first year ice

coverage, which favours the release of BrO and is widely believed to be a main driver of the BrO explosion. However, our analysis shows that there are areas where tropospheric BrO can be linked directly to locations of first year sea ice, while in other regions no clear relationship was found. Tropospheric BrO is not significantly correlated with first year sea ice extent, neither temporally nor spatially, suggesting that transport of BrO plumes away from their source regions complicates the analysis. Our comparisons hence suggest that the increase in magnitude and in the extent of the areas where BrO plumes

appear is linked in a complex way to the retreat of multi-year ice and the appearance of first year ice in the Arctic region due to Arctic Amplification. Although an overall increase of tropospheric BrO VCDs is found, there is also large inter-annual variability. While the time-series shows a positive trend on the order of 1.5% per year during polar springs, tropospheric BrO decreased in 2016. It remains to be seen if the positive trend in tropospheric BrO VCDs will continue and develop to be more pronounced in future years. Yang et al. (2010) investigated ozone loss using numerical modelling and found that

blowing snow sourced bromine reduces tropospheric ozone amounts by up to 8% in polar spring in the Arctic. Based on trends reported in the present study, future ones need to elaborate more deeply on the impact of increasing amounts of tropospheric BrO on ozone loss, mercury depletion and alteration of the oxidizing capacity of the atmosphere.

We note that the air temperature was reported to be the highest on record over the Arctic in 2016 (Overland et al., 2016). The appearance of plumes of tropospheric BrO VCDs and their intensity, are known to be influenced by meteorological drivers

(air temperature, sea level pressure, wind speeds and cyclones) and the amounts of blowing snow (Blechschmidt et al., 2016; Seo et al., 2019b). Further investigation is required in order to link the evolution of tropospheric BrO to these drivers of tropospheric BrO release, in order to better understand the relationship between them and the effect of Arctic Amplification on bromine concentrations in the Arctic region.

**Acknowledgements**

We gratefully acknowledge the funding by the Deutsche Forschungsgemeinschaft (DFG, German Research Foundation) – Projektnummer 268020496 – TRR 172, within the Transregional Collaborative Research Center "ArctiC Amplification: Climate Relevant Atmospheric and SurfaCe Processes, and Feedback Mechanisms (AC)³". We thank Francois Hendrick (BIRA-IASB) for his help on BrO data over Harestua. We thank Christian Melsheimer and the sea ice remote sensing group at the University of Bremen for their help.



**Data availability**

Part of the BrO data of this study are available through the World Data Center PANGAEA (https://doi.pangaea.de/10.1594/PANGAEA.906046). GOME2 lv1 data were provided by EUMETSAT. We acknowledge the free use of tropospheric NO$_2$ column data from GOME, GOME-2A, SCIAMACHY and GOME-2B sensors from the QA4ECV project (http://www.qa4ecv.eu/) and from the TEMIS web site (www.temis.nl). We acknowledge Mark Weber and the UVSAT group of Institute of Environmental Physics, University of Bremen, for providing total ozone columns for GOME, SCIAMACHY, GOME2-A and GOME-2B. NCEP Reanalysis data was provided by the NOAA/OAR/ESRL PSD, Boulder, Colorado, USA, through their Web site (https://www.esrl.noaa.gov/psd/). The EASE-Grid Sea ice age version 4 was provided by NSIDC from their website (https://nsidc.org/data/nsidc-0611/versions/4).

**Author contributions**

Ilias Bougoudis performed the retrieval of BrO from the different satellite instruments, collected and processed the sea ice age data, performed the analysis and wrote the paper. Anne-Marlene Blechschmidt performed the stratospheric separation. Anne-Marlene Blechschmidt, Andreas Richter, Sora Seo and John Philip Burrows provided insights and knowledge on the study. Andreas Richter developed software which was used for processing and analyzing the data. Nicolas Theys developed the stratospheric separation method. All authors contributed to the writing of the paper.

**Competing interests**

On behalf of my co-authors, I declare that I have no significant competing financial, professional or personal interests that might have influenced the performance or presentation of the work described in this manuscript.

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
