# Peer review of "Long-term Time-series of Arctic Tropospheric BrO derived from UV-VIS Satellite Remote Sensing and its Relation to First Year Sea Ice"

_Atmospheric Chemistry and Physics, 2020_

## Referee Comment (RC1) · Anonymous Referee #1 · 25 Mar 2020

This study presents a 22-year time series of Arctic tropospheric BrO from different satellite observations and discusses possible long-term changes, in particular in relation to changes in the distribution of first year sea ice. This is a pressing topic and the study is an important, highly relevant and timely contribution. As such I recommend publication in Atmos. Chem. Phys. after consideration of the general and specific comments. The manuscript is largely well written, but can be improved in parts for clarity of presentation. I have made specific comments and in addition would like to ask the senior coauthors to help improving the presentation where necessary.

General comments

[Figure]

A central hypothesis of this study is that the transition from multi-year to first year sea ice in the Arctic ocean affects the distribution, number and intensity of bromine explosion events and thus Arctic tropospheric BrO. The potential mechanism(s) behind should be made a bit clearer in the Introduction and the discussion of results should be made under these assumptions. To be more specific: Multi-year ice that has survived at least one melt cycle has less bulk salinity compared to first year sea ice. The increase in first-year sea ice could potentially favour bromine explosion events by providing more sea salt bromine. To test this hypothesis, relating first year sea ice area with (excess) tropospheric BrO, as done in Fig. 11 is very useful and the implications of this relation should be stated accordingly. The correlation between first year sea ice area and BrO seems to be significant (by looking at Fig.11, I couldn't find a statistical test for significance) although of course not perfect. How much this relation explains (in terms of variance, trend, shift in geographic distribution) should be stated clearly, together with its limitations. Phrases like "linked in a complex way" (end of abstract and in the Conclusions) without explaining what the "complex way" is hide more than they reveal.

There may be other factors how sea ice age impact on tropospheric BrO. E.g. through a reduction in sea ice thickness. So while for the point above the distinction between first year and multi-year ice is most important (and the distinction between say 5 or 6 year old ice is not so important), for other factors the actual age may be critical. Please be specific in the discussion and in the presentation. Example 1: the colour scale in Fig. 9 makes it difficult to clearly identify if the majority of the ice in recent years is first year ice or 2 year old ice. Example 2: the age trend in Fig. 13: Even a relatively small trend may have resulted in a shift from multi-year to first year ice over the years. Try to be specific in the discussion of the implications.

Section 3.1 contains a general description of the DOAS method used to retrieve the BrO SCDs. I am not sure if a general description of the DOAS method is needed here, but in its current form there are too many mistakes and omissions to make it useful (specific points below). Please carefully check.

All these points as well as the specific points below should be easily improved so that I consider the changes needed as mostly minor in terms of time needed but nevertheless important to improve the manuscript.

Specific comments

P1,L11: First sentence of abstract is a bit disconnected here. Would be better in the Introduction. More generally I feel that at many places (in the abstract and elsewhere in the manuscript) "Arctic Amplification" could better be replaced by "Arctic warming" because what matters in this context is the warming, not so much the amplification (although this may be seen as pettifoggery).

P1, L15: "e.g. Hg": any other metals?

P1, L15: "22 year": suggestion: name range of years already here

P1,L19/20: "magnitude of BrO...of about 1.5%/year": what exactly increases with 1.5%/year? Tropospheric columns? Please be specific.

P1, L25: It is true that the understanding of Arctic Amplification is inadequate, but a few citations on Arctic Amplification may be useful. E.g. Pithan and Mauritsen, Nature Geosc., 2014.

P1, L24/25: "loss of sea ice" and "reduction of ice extent" are of course not independent. Maybe say loss of ice resulting in reduction of ice extent, thickness and reduced fraction of multi-year ice?

P2, L10: "over 30 years ago"

P2, L12: introduce "O3" when first used as "ozone (O3)"

P2, L13: I think this point should be made a bit clearer: O3 and OH are decreasing, but bromine radicals instead could act as oxidising agents. Are there references how the oxidising capacity overall changes?

P2, L21: wicket sentence

P3, R4,R8: should that be "->" instead of "="?

P3, L19: Closing bracket missing.

P3, L19: This idea needs a bit more explanation: Transport of BrO plumes over large distance by deposition and reactivation due to release from snow pack and blowing snow?

P3, L24: "Polar Regions" -> "polar regions" (at many places in the manuscript)

P3, L24: "hostile" for what or whom? In spite of difficulties numerous studies have performed in-situ measurements or ground-based DOAS measurements in the Arctic. Satellite measurements are not "unique".

P3, L33: It is good to provide context by citing previous studies, but better cite what has been learned rather only what has been done.

P4, L2: "them" = "BrO explosion events" ?

P4, L10: at some stage you should mention that first year sea ice is more saline than multi-year ice

P4, L26: remove word "results"

P5, L9: This statement is too general. There were other satellite instruments before GOME, depending on what you mean by "many" and "key trace gases"

P5, L16: what is the difference between "near IR" and "short wave IR"?

P6, L5: throughput mentioned twice

P6, L9: This sentence seems odd. "Long changing polymers" -> "long chained polymers"?

P7, Eq.1: In eq.(1) the concentration of the gas j is missing.

P7, L14: really ABSORPTION cross sections or SCATTERING cross sections?

P7, L31: is a "four degree" polynomial a polynomial of fourth-order, i.e. with five degrees of freedom? Please be specific and consistent to avoid confusion.

P8, Eq.5: There is something wrong with eq. (5). The sum should be under the root. And not on the LHS.

P8, L16: Latitudes and longitudes need to be specified more consistently. E.g. "-180°E" should be "180°W" and "-50.0°S" should be "50°S". If you use east and west longitudes please avoid "235°E to 270°E" and use instead "90°W to 125°W".

P10, L1: Please mention this drift already in the instrument overview in Section 2.

P11, L2: This first sentence does not make much sense and is redundant.

P13, L13: what is the meaning of the word "high" here?

P13, L15: you can remove "due to Arctic Amplification" here. See also my general comment on Arctic Amplification.

P14, L10: There are also real stratospheric BrO trends due to changes in anthropogenic emissions!

P15, L1: I don't think you need to explain that temperatures in summer are higher because of increased solar insolation (it does not matter in this context), but you could specify in which month the maximum temperature and in which month the sea ice minimum are reached.

P15, L7: Looking at the spatial distribution may tell if there are (few and/or small) areas of bromine explosion in September or only a gradual increase in the background. Would it make sense to include somewhere also maps with BrO in autumn?

P22, L4: "in the early years, most of the BrO is found in the region of the Barents and Kara Sea": I am confused. Richter et al. (1998) show largest BrO enhancements in

March 1997 over the Canadian Archipelago, Hudson Bay and north-west of Greenland.

P24, Fig11: I don't fully understand what are the points shown in the right hand side panels? Related to this: what is the correlation coefficients for the data shown in the time series on the left hand side? The same as given in the right hand side plots? Did you check if the correlations are significant? (By eye, the correlation between Arctic MAM mean BrO and 1st year ice extent in Fig. 11a seems significant.)

P24, L11: the phrase "fresh" ice is misleading, as first year ice has a higher salinity, i.e. is not "fresh"

P24, L15: What is the correlation coefficient? (See comment above.) A rigorous statistical analysis is more useful than the "anecdotal evidence" given in the following sentences.

P25, L4-7: I find these sentences confusing. My impression from Fig. 11a is that there is a positive correlation between first year sea ice and BrO, although this is clearly not the only factor. But then you say this agrees with Choi et al. who found even a negative correlation? I would say this is in contrast to Choi et al. and part of this difference may be attributed to a possible degradation of the OMI instrument?

P25, L22: Please specify the value of the autocorrelation used. What exactly is the meaning of the "period M"? Generally I am not convinced that the trend model with harmonics is an ideal choice as you don't have data during winter. For calculating trends in individual months as in Table 4 the harmonics are not needed at all.

P25, Table 4: It is a bit unusual to have the units in a column, instead of in the header.

P28, L10: First paragraph of Summary is redundant, largely repeats introduction

P29, L7: did you show that the correlation in Fig. 11a is not significant?

P29, L18: This statement on temperatures in 2016 is too vague. You could mention here that you have not considered changes in temperature and this may be another

factor affecting tropospheric BrO to be investigated in future studies.

P30, Author contributions: Sounds a bit strange that not all authors are named for their contributions here (not even for "providing insight and knowledge").

P31: Barrie and Platt listed twice

P32: Claas listed twice.

P32, L23: something is missing here

P33: Fickert listed twice

---

## Referee Comment (RC2) · Anonymous Referee #2 · 17 Jun 2020

Review of 'Long-term Time-series of Arctic Tropospheric BrO derived from UV-VIS Satellite Remote Sensing and its Relation to First Year Sea Ice' Bougoudis et al.,

The paper is well written overall. However, there lacks clear take home messages, i.e. how does this work significantly advance our knowledge. I would like to come away with more than the relationship is 'complex'. I would like quantification of the first year sea-ice extent and Arctic amplification on BrO explosion events as these have been nicely explored in this paper, but the clarity of conclusions drawn reduces the potential significance and usefulness of this work. Below I detail edits throughout the paper with my major concern being around the significance of this work, and how it could be used

to advance model, observation comparisons going forward, so that predictions of the implications for the oxidative capacity of the Arctic (and therefore also the Antarctic) can be explored in climate simulations.

Page 1, line 12: Sentence needs revising: Every polar spring BrO explosions occur, which are a series of chemical reactions that release bromine molecules to the troposphere over sea ice covered regions.

Page 1, line 16 Name the four satellite experiments

Page 1, line 20 and elsewhere: 1.5% per year – since there is a focus in the abstract on Arctic Amplification – could this response be expressed also per degree of warming experienced?

Page 1, line 22 Specify how linked (stating that it is complex is very general)

Page 1, line 24 Arctic temperature please qualify as surface air or sea-surface temperature?

Page 2, line 6 here and elsewhere please remove hyphen in ecosystem

Page 2, line 11 1990 and 1997 don't represent 3 decades worth of studies, a 2010s review paper reference more appropriate here

Page 2, line 14 insert but . . . OH are reduced, but the reactions of. . .

Page 2, line 17 the "Bromine explosion" closing speech mark needs format correction

Page 2, line 20 citing a 2009 paper when discussing controversy is problematic, as modelling efforts recently have integrated simple and effective parameterisations, based on both frost flower and blowing snow mechanisms i.e. (Falk & Sinnhuber, 2018) or some other more recent reference would be appropriate here to discuss exactly what controversary still exists.

R1 – hv should be hïĄőïĂăïĂĺthese should be defined in the text)

[Figure]

Page 3 line 7 bromine atoms rapidly remove (switch from remove rapidly)

Page 3 line 20 it's = its

Page 3, line 23 favor bromine explosion conditions. Again, there is more recent (and quantitative) modelling efforts.

Page 3, line 17 remove second comma

Page 5, line 16

Page 5, line 23 on – was the data from Metop-C useable in theory, if so why wasn't it used in your study?

Table 1 in GOME-2A line 40 x 40 (remove capitalisation from X)

Page 6, line 9 and elsewhere – Sun should be capitalised throughout

Page 9, line 5 insert comma: autumn, when the solar. . .

Page 11, line 2 (whole paragraph needs reworking): For the NO2 and O3 column satellite retrievals, the tropopause height . . . is used. . ..

Page 11 Line 4: reflectivity, which is required. . .

Page 13, line 11 change the degree sign from a zero to a circle

Figure 5 – please show 1 sigma errors in the VCD retrievals as well as interannual variability with error bars on these plots in order for us to be able to determine agreement significance.

Section 4.2, by selecting only for high BrO over sea-ice areas you are only capturing the genesis and not necessarily determining the implications of the combined oxidative capacity and changes in cyclonic activity. Much of the interesting implications of warming and BrO will happen outside of areas which are over sea-ice. I'm unclear why the only BrO over sea-ice was a necessary criteria?

Figure 9 – please try to get onto one page, use horizontal colorbars perhaps?

Page 22, line 16. And Figure 11 Stating that the relationship between sea-ice area and BrO explosion is complex is unsubstantiated by a useful plot. I would like to see the first year sea-ice area versus average BrO column plot (or BrO explosion area). Figure 11d goes some of the way to providing this but I suggest that an annual number be provided i.e. those produced in 11 a and b (with variability range errorbars for both axes) could be more useful than using every data point above the threshold as is done currently. Figure 11d trend line is arbitrary and should be removed (unless the annual number provides support for a linear relationship). Some metric representation of annual cyclonic activity or amplification amplitude would be valuable too on this plot – i.e. providing an annual cyclonic index in color and some index for Arctic amplification temperature with size may help to disentangle the story for us. This way we would be able to determine whether a parameterization that would be useful to test models could be derived for this 'complex' relationship.

Page 25, line 4 VD -> VCD

Figure 11 a-d labels missing

Page 27, figure 12. How can trends over 1 year be considered significant or reliable with only a few points? I assume the annual cycle is removed for this plot (otherwise like a sine curve, you would get a trend just due to that). I'm really unclear about what figure 12 is showing, given it only is discussed briefly and doesn't add to the papers aims/conclusions. As the minimum time to detect a trend is not provided, I think this plot and the discussion can be removed.

Figure 13, over what time period is the trend calculated? Provide this in the caption.

Page 29 line 5 the BrO explosion -> Bro explosion events.

Page 29 when discussing trends state over what period they are determined.

---

## Author Comment (AC1) · 8 Aug 2020

We would like to thank the reviewer for providing helpful comments and suggestions for improvement of the paper. We believe that they enhance the scientific value of our contribution. In the following, we discuss the questions and suggestions and give a point by point response.

**General comments**

*A central hypothesis of this study is that the transition from multi-year to first year sea ice in the Arctic ocean affects the distribution, number and intensity of bromine explosion events and thus Arctic tropospheric BrO. The potential mechanism(s) behind should be made a bit clearer in the Introduction and the discussion of results should be made under these assumptions. To be more specific: Multi-year ice that has survived at least one melt cycle has less bulk salinity compared to first year sea ice. The increase in first-year sea ice could potentially favour bromine explosion events by providing more sea salt bromine.*

*To test this hypothesis, relating first year sea ice area with (excess) tropospheric BrO, as done in Fig. 11 is very useful and the implications of this relation should be stated accordingly. The correlation between first year sea ice area and BrO seems to be significant (by looking at Fig.11, I couldn't find a statistical test for significance) although of course not perfect. How much this relation explains (in terms of variance, trend, shift in geographic distribution) should be stated clearly, together with its limitations. Phrases like "linked in a complex way" (end of abstract and in the Conclusions) without explaining what the "complex way" is hide more than they reveal.*

*There may be other factors how sea ice age impact on tropospheric BrO e.g. through a reduction in sea ice thickness. So while for the point above the distinction between first year and multi-year ice is most important (and the distinction between say 5 or 6 year old ice is not so important), for other factors the actual age may be critical. Please be specific in the discussion and in the presentation. Example 1: the colour scale in Fig. 9 makes it difficult to clearly identify if the majority of the ice in recent years is first year ice or 2 year old ice. Example 2: the age trend in Fig. 13: Even a relatively small trend may have resulted in a shift from multi-year to first year ice over the years. Try to be specific in the discussion of the implications.*

Currently, although the Arctic sea ice extent is decreasing, the more saline first year ice grows at the expense of multiyear ice. This is affecting the number and magnitude of bromine explosions occurring. The criticism is that there is no clear conclusion on the relationship between tropospheric BrO and first year ice conditions. In response your comment, we have rephrased parts of the abstract, the introduction, the main text and the summary, (e.g. the moderate correlation coefficient of 0.32 between the two quantities during polar springs is now given on the abstract). With the help of maps in Fig. 9 and time-series in Fig. 11, we have pointed out that changing sea ice conditions do not explain all the changes of the observed tropospheric BrO. This is because the appearance of bromine explosions depends on other parameters as well (air temperature, wind speeds, cyclone activity).

We have performed a hypothesis test of the significance of the correlation coefficient as you suggested, based on the null hypothesis that there is no correlation between tropospheric BrO

and first year ice. The null hypothesis was rejected, meaning that the correlation between tropospheric BrO and first year ice is significant. The color scale in Fig. 9 was optimized in a way that first year ice is distinguishable from multiyear ice. Values from 1 to 2 are colored blue, while multi-year ice (2 years and above) is denoted with green and other colors. So everything blue in the maps of Figure 9 means that there was at least one first year ice grid box during the polar spring average. Also, in Fig. 13, we tried to explain better the point, which you have made by adding another sub-plot in the figure, which shows the trend of the occurrences of first year ice. We infer from it that the trend of occurrences of first year ice has increased in some areas where also an increase of tropospheric BrO occurred (i.e. east of Greenland), but also decreased in some others (i.e. in the Canadian Archipelago, where BrO also increased). In summary, we explain that sea ice age influences tropospheric BrO but, given that the correlations between first year ice and daily tropospheric BrO are moderate, does not fully account for the changes of the tropospheric BrO column dataset.

*Section 3.1 contains a general description of the DOAS method used to retrieve the BrO SCDs. I am not sure if a general description of the DOAS method is needed here, but in its current form there are too many mistakes and omissions to make it useful (specific points below). Please carefully check.*

With respect to your comments for the DOAS section, we have followed the specific comments (indicated below):

**Specific comments**

*P1,L11: First sentence of abstract is a bit disconnected here. Would be better in the Introduction. More generally I feel that at many places (in the abstract and elsewhere in the manuscript) "Arctic Amplification" could better be replaced by "Arctic warming" because what matters in this context is the warming, not so much the amplification (although this may be seen as pettifoggery).*

We removed the first sentence. We have also replaced in all places (except in the introduction where we describe the term "Arctic Amplification") the term "Arctic Amplification" with "Arctic warming".

*P1, L15: "e.g. Hg": any other metals?*

Bromine molecules can also react with lead, forming $PbBr_2$. However $PbBr_2$ is insoluble.

*P1, L15: "22 year": suggestion: name range of years already here*

Rephrased to: "22 year (1996 to 2017)"

*P1,L19/20: "magnitude of BrO: : :of about 1.5%/year": what exactly increases with 1.5%/year? Tropospheric columns? Please be specific.*

Rephrased to: "We determined an increasing trend of about 1.5% of the tropospheric BrO VCDs per year during polar spring".

*P1, L25: It is true that the understanding of Arctic Amplification is inadequate, but a few citations on Arctic Amplification may be useful. E.g. Pithan and Mauritsen, Nature Geosc., 2014.*

Added two more references (Pithan and Mauritsen, 2014, Stjern et al, 2019), which discuss the response of Arctic Amplification to individual climate drivers and to temperature feedbacks.

*P1, L24/25: "loss of sea ice" and "reduction of ice extent" are of course not independent. Maybe say loss of ice resulting in reduction of ice extent, thickness and reduced fraction of multi-year ice?*

Corrected the sentence to: "… are the loss of ice, resulting in reduction of ice extent, thickness and a reduced fraction of multi-year ice (Stroeve et al., 2012) and the increasing rate of loss of the Greenland ice cap (Mouginot et al., 2019)".

*P2, L10: "over 30 years ago"*

Added

*P2, L12: introduce "O3" when first used as "ozone (O3)"*

Done

*P2, L13: I think this point should be made a bit clearer: O3 and OH are decreasing, but bromine radicals instead could act as oxidising agents. Are there references how the oxidising capacity overall changes?*

Indeed, on short timescales, bromine radicals can contribute to the formation of OH. However, on longer timescales, the depletion of ozone caused by bromine compounds reduces the production of OH. Added a reference (Stone et al, 2018), which suggests that the long-term effect of bromine radicals prevails. Also, added one reference on the global tropospheric OH distribution (Lelieveld et al., 2016), which suggests that the secondary sources of OH (i.e. recycling in radical reaction chains), play a more important role in global OH distribution.

*P2, L21: wicket sentence*

Rephrased to: "However, there is the general consensus that the potential sources of BrO plumes are (a) rich in sea salts and relatively cold (conditions occurring in potential frost flowers regions; Rankin et al., 2002; Kaleschke et al., 2004; Sander et al., 2006), (b) surfaces covered with liquid or frozen brine (Sander et al., 2006), (c) associated with blowing snow (Yang et al., 2008; Blechschmidt et al., 2016; Frey et al., 2019), (d) surface snow packs (Pratt et al., 2013; Peterson et al., 2018) and young salty sea ice regions (Wagner et al., 2001; Simpson et al., 2007; Peterson et al., 2016)."

*P3, R4,R8: should that be "->" instead of "="?*

Changed to $\rightleftharpoons$

*P3, L19: Closing bracket missing.*

Added

*P3, L19: This idea needs a bit more explanation: Transport of BrO plumes over large distance by deposition and reactivation due to release from snow pack and blowing snow?*

Rephrased to: "It was shown that BrO plumes can be transported far from their initial formation areas, as high wind speeds associated with cyclones (Begoin et al., 2010; Zhao et al., 2015; Blechschmidt et al., 2016) can transfer them together with blowing snow (Giordano et al., 2018)".

*P3, L24: "Polar Regions" -> "polar regions" (at many places in the manuscript)*

Replaced by "polar regions" as suggested

*P3, L24: "hostile" for what or whom? In spite of difficulties numerous studies have performed in-situ measurements or ground-based DOAS measurements in the Arctic. Satellite measurements are not "unique".*

Changed the sentences containing the problematic words to: "The polar regions are some of the most remote places on the planet. Consequently, satellite remote sensing is a suitable method to study bromine chemistry in the Arctic."

*P3, L33: It is good to provide context by citing previous studies, but better cite what has been learned rather only what has been done.*

We have rephrased the whole section, now mentioning also the findings of each publication: "The relationship between BrO release and young sea ice was also discussed (Wagner et al., 2001), where it was indicated that large BrO concentrations are found over or near sea ice on the Caspian Sea. Van Roozendael et al. (2004) compared SCIAMACHY observations of Arctic BrO to GOME data, showing satisfactory agreement between the two different sensors. Theys et al. (2011) compared tropospheric BrO columns derived from GOME-2A to a chemical transport model, showing consistency with the release mechanisms of bromine. Sihler et al. (2012) compared GOME-2 BrO columns to ground-based measurements in the Arctic, demonstrating good agreement between the retrievals. Seo et al. (2019a) presented the first BrO retrievals from the TROPOMI instrument, showing high-resolution bromine explosion cases with low fitting errors".

*P4, L2: "them" = "BrO explosion events" ?*

Replaced as suggested

*P4, L10: at some stage you should mention that first year sea ice is more saline than multi-year ice*

Rephrased the sentence to: "Changes in meteorological parameters, e.g. increasing air temperature (Serreze and Barry, 2011), decreasing mean sea level pressure over northeastern America and increasing pressure over Eurasia (Ogawa et al., 2018; McCusker et al., 2016), increase in cyclone frequency and intensity (Akperov et al., 2019), stronger surface winds (Mioduszewski et al., 2018) and changes in sea ice conditions (e.g. reduced sea ice extent; Stroeve et al., 2012), increased first year sea ice fraction (and consequently salinity), and therefore decreased sea ice thickness (Richter-Menge et al., 2017) occur due to Arctic warming.". Also included one sentence in the introduction: "The reduced multi-year ice is being replaced by first year ice, which is in addition more saline. (Galley et al., 2016)" (P1, L30).

*P4, L26: remove word "results"*

Removed

*P5, L9: This statement is too general. There were other satellite instruments before GOME, depending on what you mean by "many" and "key trace gases"*

Rephrased the sentence to: "GOME was the first satellite instrument, which was able to measure key tropospheric gases which have weaker absorption lines than ozone: examples are NO2, BrO, HCHO and SO2".

*P5, L16: what is the difference between "near IR" and "short wave IR"?*

Although the spectral limits of the terms near IR and short wave IR are not SI standardized, near-infrared radiation usually implies a spectral wavelength region from 0.75 to 1.4 μm, while short-wavelength infrared implies the spectral region from 1.4 to 3 μm. Shortwave IR is sometimes defined as 0.76 to 2 μm. The sentence has been changed to: "... allowing the observation of many trace gases in the near infrared (0.75 to 1.4 μm) and short wave infrared (1.4 to 3 μm) spectral wavelength regions".

*P6, L5: throughput mentioned twice*

Corrected

*P6, L9: This sentence seems odd. "Long changing polymers" -> "long chained polymers"?*

Changed to "long chain polymers"

*P7, Eq.1: In eq.(1) the concentration of the gas j is missing.*

We define the concentration in this study as the number density of the trace gas j with concentration $\rho$ (in molec cm$^{-3}$). This, when multiplied by the length of the light path s (in

meters), gives the column amount or column (in molec cm$^{-2}$) of the trace gas j. We have rewritten Equation 1 as follows:

$$I = I_o \, e^{-\int \Sigma_{j=1}^{J}\{\sigma_j(\lambda)\rho_j\}ds}$$

Also, we have rephrased the text below to: "where I is the measured intensity of the electromagnetic radiation, $I_o$ is the initial intensity, J is the total number of absorbing trace gases, j denotes a particular trace gas (e.g. BrO), $\sigma(\lambda)$ is the cross section of the absorber at wavelength $\lambda$, $\rho$ the concentration of the trace gas".

*P7, L14: really ABSORPTION cross sections or SCATTERING cross sections?*

Changed to scattering cross sections

*P7, L31: is a "four degree" polynomial a polynomial of fourth-order, i.e. with five degrees of freedom? Please be specific and consistent to avoid confusion.*

Changed to fourth-order polynomial

*P8, Eq.5: There is something wrong with eq. (5). The sum should be under the root. And not on the LHS.*

Corrected

*P8, L16: Latitudes and longitudes need to be specified more consistently. E.g. "- 180_E" should be "180_W" and "-50.0_S" should be "50_S". If you use east and west longitudes please avoid "235_E to 270_E" and use instead "90_W to 125_W".*

Changed to "the Arctic from 70.0$^o$ N to 85.0$^o$ N latitude, 180$^o$ W to 180$^o$ E longitudes) and for the Pacific reference region (50.0$^o$ S to 10.0$^o$ N latitude and 90.0$^o$ W to 125.0$^o$ W longitude).".

*P10, L1: Please mention this drift already in the instrument overview in Section 2.*

Moved to section 2.4

*P11, L2: This first sentence does not make much sense and is redundant.*

The sentence has been changed to: "NO$_2$ and O$_3$ columns from satellite retrievals and tropopause height from meteorological reanalysis data are used for deriving the tropospheric BrO component from the retrieval (stratospheric separation)."

*P13, L13: what is the meaning of the word "high" here?*

Removed from the sentence

*P13, L15: you can remove "due to Arctic Amplification" here. See also my general comment on Arctic Amplification.*

Removed

*P14, L10: There are also real stratospheric BrO trends due to changes in anthropogenic emissions!*

The sentence has been changed to: "The stratospheric BrO VCDs show a small upward trend from 1995 to 2001 and a slight decrease afterwards, which are in agreement with measurements of stratospheric BrO from Harestua station (Hendrick et al., 2008)."

*P15, L1: I don't think you need to explain that temperatures in summer are higher because of increased solar insolation (it does not matter in this context), but you could specify in which month the maximum temperature and in which month the sea ice minimum are reached.*
Changed to: "During July and August, the temperature reaches its maximum. In September, the minimum sea ice extent is observed."

*P15, L7: Looking at the spatial distribution may tell if there are (few and/or small) areas of bromine explosion in September or only a gradual increase in the background. Would it make sense to include somewhere also maps with BrO in autumn?*

During September, the sea ice coverage is very low, so it would not make sense to include September maps of tropospheric BrO in the manuscript. However, we have checked September maps of tropospheric BrO and concluded that the small increase observed is mainly a background increase, as we cannot see clearly localized BrO explosion events.

*P22, L4: "in the early years, most of the BrO is found in the region of the Barents and Kara Sea": I am confused. Richter et al. (1998) show largest BrO enhancements in March 1997 over the Canadian Archipelago, Hudson Bay and north-west of Greenland.*

Richter et al showed specific days of total BrO vertical column densities. Although the Hudson Bay is a well known BrO hotspot, we decided to perform our analysis only over latitudes $> 70^0$ N, as the proportion of sea ice covered areas and the fraction of pixels for which tropospheric BrO VCDs can be retrieved is larger there. Retrieved tropospheric BrO columns are more accurate for latitudes $> 70^0$ N, as the tropospheric air mass factor that we use considers a 0.9 surface reflectivity.
We have analyzed retrievals over Hudson Bay, and we are attaching figures of geometric and tropospheric BrO VCDs (similar to Figure 3 of the paper, with a land mask applied):

[Figure]

We see a less pronounced trend for BrO VCDs over the Hudson Bay, in comparison with the Arctic region, as defined in the manuscript (latitudes > 70$^{o}$)

*P24, Fig11: I don't fully understand what are the points shown in the right hand side panels? Related to this: what is the correlation coefficients for the data shown in the time series on the*

*left hand side? The same as given in the right hand side plots? Did you check if the correlations are significant? (By eye, the correlation between Arctic MAM mean BrO and 1st year ice extent in Fig. 11a seems significant.)*

Every point in the right panel scatter plots represents one polar spring day (the average tropospheric BrO VCD for this day is plotted against first year sea ice extent in the top right panel, the area with tropospheric BrO VCD above the threshold is plotted against first year sea ice extent in the bottom right). The time-series on the left side are the polar spring averages of the scatter plots. The correlation is positive but moderate for the top right scatter plot. We have performed a hypothesis test to verify the significance of the correlation coefficient and found that the correlation is significant (p value lower than 0.05). We added this information to the text of the paper. Also, we have calculated the correlation coefficient of the annual time-series (the blue and red curve shown in the left sub-figures). The correlation coefficient between polar spring averages of tropospheric BrO and first year ice extent corresponding to the top left sub-figure is 0.62, while the correlation between areas of polar spring averages of tropospheric BrO VCDs > 7 x $10^{13}$ molec cm$^{-2}$ and first year ice is 0.46. However, we believe that the correlation coefficients obtained from the daily scatter plots are more meaningful and represent the actual relationship of tropospheric BrO and first year ice extent more accurately.

*P24, L11: the phrase "fresh" ice is misleading, as first year ice has a higher salinity, i.e.is not "fresh"*

Changed to "and more first or young sea ice is formed each winter period"

*P24, L15: What is the correlation coefficient? (See comment above.) A rigorous statistical analysis is more useful than the "anecdotal evidence" given in the following sentences.*

The correlation coefficients for the two scatter plots are presented inside the plots (denoted with r). The results of the significance test has been added to the text.

*P25, L4-7: I find these sentences confusing. My impression from Fig. 11a is that there is a positive correlation between first year sea ice and BrO, although this is clearly not the only factor. But then you say this agrees with Choi et al. who found even a negative correlation? I would say this is in contrast to Choi et al. and part of this difference may be attributed to a possible degradation of the OMI instrument?*

Although the correlation between tropospheric BrO and first year ice is positive from the scatter plot, we see that it is not strong, as many days with high tropospheric BrO columns occur with moderate first year ice extent (around 4.5x$10^6$ km$^2$). From the time-series, we see in many years an opposite evolution (one quantity increases, while the other decreases, for example 2008 and 2015). The text was rephrased, especially in the section where we compare to Choi et al, to: "This finding is in contrast to the results by Choi et al. (2018), where an analysis of BrO VCD retrieved from OMI was performed. They found a correlation coefficient of -0.32 between first year ice extent and tropospheric bromine explosion frequency. The negative sign is attributed to a decrease of tropospheric BrO VCDs over the latter years of the trend analysis. The differences

to Choi et al. (2018) may be explained by a degradation of the OMI instrument (Kroon et al., 2011).".

*P25, L22: Please specify the value of the autocorrelation used. What exactly is the meaning of the "period M"? Generally I am not convinced that the trend model with harmonics is an ideal choice as you don't have data during winter. For calculating trends in individual months as in Table 4 the harmonics are not needed at all.*

An autocorrelation value of 0.2 was used. The period M is the 22 years of the entire dataset. The reason why we used this approach is that, although the winter columns are missing, we still see a profound seasonality in our dataset. However, when we calculated the monthly trends in table 4, this approach was not used (added in the text).

*P25, Table 4: It is a bit unusual to have the units in a column, instead of in the header.*

Added the unit information to the title and in the header of the figure

*P28, L10: First paragraph of Summary is redundant, largely repeats introduction*

Rephrased – removed most parts of the first paragraph

*P29, L7: did you show that the correlation in Fig. 11a is not significant?*

We have performed a hypothesis test on the significance of the correlation coefficient. The correlation is significant and we have added it in the text.

*P29, L18: This statement on temperatures in 2016 is too vague. You could mention here that you have not considered changes in temperature and this may be another factor affecting tropospheric BrO to be investigated in future studies.*

Rephrased to "The appearance of plumes of tropospheric BrO VCD and their intensity are influenced by several meteorological drivers (air temperature, sea level pressure, wind speeds and cyclones) and the amounts of blowing snow (Blechschmidt et al., 2016; Seo et al., 2019b). Further investigations are required to understand the evolution of tropospheric BrO and its dependence on these drivers of tropospheric BrO release."

*P30, Author contributions: Sounds a bit strange that not all authors are named for their contributions here (not even for "providing insight and knowledge").*

Rephrased to: "I. Bougoudis undertook the retrieval of BrO from the different satellite instruments, collected and processed the sea ice age data, performed the analysis and prepared the paper. This study was initiated by J.P. Burrows and A.-M. Blechschmidt. The research presented was supervised by A.-M. Blechschmidt, A. Richter and J.P. Burrows. A.-M. Blechschmidt and N. Theys provided the stratospheric separation. A.-M. Blechschmidt, A. Richter, S. Seo, N. Theys and J.P. Burrows provided input with respect to BrO issues of relevance. A. Richter developed software which was used for processing and analysing the BrO

data. A. Rinke provided input on sea ice and trend analyses. All authors contributed to the writing of the paper.".

*P31: Barrie and Platt listed twice*

Corrected

*P32: Claas listed twice.*

Corrected.

*P32, L23: something is missing here*

Inserted the missing reference

*P33: Fickert listed twice*

Corrected

---

## Author Comment (AC2) · 8 Aug 2020

We thank the reviewer for providing the helpful and critical comments and suggestions about the paper. We believe that they enhance the scientific value and accuracy of the manuscript. In the following, we discuss the questions and suggestions and provide point by point responses.

**General comment**

*"The paper is well written overall. However, there lacks clear take home messages, i.e. how does this work significantly advance our knowledge. I would like to come away with more than the relationship is 'complex'. I would like quantification of the first year sea-ice extent and Arctic amplification on BrO explosion events as these have been nicely explored in this paper, but the clarity of conclusions drawn reduces the potential significance and usefulness of this work. Below I detail edits throughout the paper with my major concern being around the significance of this work, and how it could be used to advance model, observation comparisons going forward, so that predictions of the implications for the oxidative capacity of the Arctic (and therefore also the Antarctic) can be explored in climate simulations".*

We have re-written the abstract, introduction and conclusion sections, in order to clearly state the importance of the current study and removed the general statement that the relation between tropospheric BrO VCDs and sea ice is complex. We observe changes in tropospheric BrO plumes in recent years. These changes can be linked non-linearly to changes in sea ice age and extent that are also occurring. This is now pointed out in the manuscript. We provide the correlation coefficient between first year ice extent and tropospheric BrO in the abstract, in order to clarify and quantify the relation from the beginning. Furthermore, we have added a significance test between the two quantities, in order to clarify, that although their correlation (based on the magnitude of the correlation coefficient) is moderate, it is significant. Also, we have included another subplot in Figure 13, where we show the trend of first year ice occurrences. In this way, our conclusions on the impact of first year ice on the recent increased formations of tropospheric BrO plumes are more solid. In the conclusion, we have erased the repetitions and kept only the take-home messages: The 1.5% increase of tropospheric BrO during polar springs and its moderate correlation of 0.32 to the first year ice extent evolution. Also, we discuss potential future significance and usage of work. The dataset that we retrieved can be integrated in chemical transport models, in order to be used as validation of simulations on the impact of bromine explosions on $O_3$ loss and potential OH changes.

With respect to your detailed edits, we have followed the specific comments (indicated below):

**Specific comments**

*Page 1, line 12: Sentence needs revising: Every polar spring BrO explosions occur, which are a series of chemical reactions that release bromine molecules to the troposphere over sea ice covered regions.*

Thank you for pointing to this sentence. We rephrased the sentence to ". Every polar spring, phenomena called bromine explosions occur over sea ice. These bromine explosions comprise photochemical heterogeneous chain reactions that release bromine molecules, Br2, to the troposphere and lead to tropospheric plumes of bromine monoxide, BrO."

*Page 1, line 16: Name the four satellite experiments*

Added (GOME, SCIAMACHY, GOME-2A and GOME-2B)

*Page 1, line 20: and elsewhere: 1.5% per year – since there is a focus in the abstract on Arctic Amplification – could this response be expressed also per degree of warming experienced?*

This is an interesting suggestion. However, the non-linear connection of tropospheric BrO plumes with their driving mechanisms (as also indicated in the paper, the comparisons of tropospheric BrO and sea ice age can be seen in some cases, but not in all), and the deep interactions between the mechanisms themselves, make the specification of the effect of the warming of the air temperature on bromine explosions and the trends appearing in the time-series a difficult task. We therefore prefer not to add a statement such as "BrO increased by 5% per K warming" as this would suggest a linear relationship, which in our opinion does not exist. However, we are attaching here the time-series of tropospheric BrO and 2m Arctic air temperature, from 2 reanalysis datasets (ECMWF ERA-5 (Hersbach, H., Bell, B., Berrisford, P., Hirahara, S., Horányi, A., Muñoz-Sabater, J., et al. (2020). The ERA5 global reanalysis. Quarterly Journal of the Royal Meteorological Society, 146(730), 1999–2049. https://doi.org/10.1002/qj.3803), downloaded from the following website: https://www.ecmwf.int/en/forecasts/datasets/reanalysis-datasets/era5, and WRF Arctic System Reanalysis version 2 (Bromwich, D. H., Wilson, A. B., Bai, L., Liu, Z., Barlage, M., Shih, C.-F., et al. (2018). The Arctic System Reanalysis, Version 2. Bulletin of the American Meteorological Society, 99(4), 805–828. https://doi.org/10.1175/BAMS-D-16-0215.1), downloaded from: https://rda.ucar.edu/datasets/ds631.1/).

[Figure]

*Page 1, line 22: Specify how linked (stating that it is complex is very general)*

We have rephrased the text, stating that the link (based on the magnitude of the correlation coefficient) is moderate. "We infer from comparisons and correlations with sea ice age data that the reported changes in the extent and magnitude of tropospheric BrO VCDs are moderately related to the increase of first-year ice extent in the Arctic north of 70$^o$ N, with a correlation coefficient of 0.32.".

*Page 1, line 24: Arctic temperature please qualify as surface air or sea-surface temperature?*

Changed it to: "surface air temperature"

*Page 2, line 6: here and elsewhere please remove hyphen in ecosystem*

Removed it throughout the text

*Page 2, line 11: 1990 and 1997 don't represent 3 decades worth of studies, a 2010s review paper reference more appropriate here*

Added a reference by Saiz-Lopez and von Glasow, 2012

*Page 2, line 14: insert but : : : OH are reduced, but the reactions of: : :*

Inserted as suggested

*Page 2, line 17: the "Bromine explosion" closing speech mark needs format correction*

Corrected

*Page 2, line 20: citing a 2009 paper when discussing controversy is problematic, as modelling efforts recently have integrated simple and effective parameterisations, based on both frost flower and blowing snow mechanisms i.e. (Falk & Sinnhuber, 2018) or some other more recent reference would be appropriate here to discuss exactly what controversary still exists.*

The exact level of impact of each parameter (such as blowing snow, wind speeds, air temperature) on the formation of enhanced BrO plumes is still unknown. This is what we meant with controversy. Added citation to Falk & Sinnhuber (2018), Huang et al. (2020), Seo et al. (2019). We have rephrased the text to: "Although there are studies which try to model BrO plumes from their driving mechanisms (Falk and Sinnhuber, 2018; Seo et al., 2019b; Huang et al., 2020), the exact level of impact of each parameter on the formation of enhanced tropospheric BrO is uncertain.

*R1 – hv should be hï ̧ AʺoïˇA ˘ aïˇAʹlthese should be defined in the text).*

We are sorry but we could not figure out exactly what the reviewer wrote. However, we have defined hv as solar radiation in the text. All the other species in the chemical reactions are described in the section below the reactions.

*Page 3 line 7: bromine atoms rapidly remove (switch from remove rapidly)*

Rephrased as suggested

*Page 3 line 20: it's = its*

Corrected

*Page 3, line 23: favor bromine explosion conditions. Again, there is more recent (and quantitative) modelling efforts.*

Here we are mostly presenting studies which indicate the relation of tropospheric BrO to driving mechanisms, not modeling efforts. We added citations from Yang et al, (2020) and Fernandez et al, (2019), regarding recent efforts on BrO modeling.

*Page 3, line 17: remove second comma*

Removed (Page 4, line 17)

*Page 5, line 23: on – was the data from Metop-C useable in theory, if so why wasn't it used in your study?*

Metop-C was launched in autumn 2018, but the first data were received during spring of 2019. It therefore does not contribute to the investigated time period (1996 to 2017).

*Page 6, Table 1: in GOME-2A line 40 x 40 (remove capitalisation from X)*

Removed

*Page 6, line 9: and elsewhere – Sun should be capitalised throughout*

Changed throughout the paper

*Page 9, line 5: insert comma: autumn, when the solar: : :*

Inserted

*Page 11, line 2 (whole paragraph needs reworking): For the NO2 and O3 column satellite retrievals, the tropopause height : : : is used: : :.*

This has been rephrased to "$NO_2$ and $O_3$ columns from satellite retrievals and tropopause height from meteorological reanalysis data are used for extracting the tropospheric BrO component from the retrieval (stratospheric separation). Sea ice data (age and type) obtained by satellite remote sensing was used in order to identify regions with sea ice cover and hence high surface reflectivity, which is required for the retrieval of tropospheric BrO in this tudy and for data interpretation in relation to bromine sources."

*Page 11 Line 4: reflectivity, which is required: : :*

Corrected

*Page 13, line 11: change the degree sign from a zero to a circle*

Changed throughout the whole text

*Figure 5: – please show 1 sigma errors in the VCD retrievals as well as interannual variability with error bars on these plots in order for us to be able to determine agreement significance.*

The purpose of Figure 5 is to provide seasonal cycles for geometric, stratospheric and tropospheric BrO time-series, for each instrument. For each sensor we averaged the data over the individual months of the corresponding operation period. Therefore, and since we can see from Figure 3 that for some years we have higher columns than for others (e.g. 2013 to 2017, the GOME-2B operation period compared to 1996 to 2003, the GOME operating period), the agreement between the sensors is hence not the focus of Figure 5 and differences because of the different averaging time periods are expected.

*Section 4.2: by selecting only for high BrO over sea-ice areas you are only capturing the genesis and not necessarily determining the implications of the combined oxidative capacity and changes in cyclonic activity. Much of the interesting implications of warming and BrO will happen outside of areas which are over sea-ice. I'm unclear why the only BrO over sea-ice was a necessary criteria?*

We chose to work exclusively on sea ice covered regions because the tropospheric air mass factor that we use in the computation of tropospheric columns considers a surface reflectivity of 0.9. As a result, our tropospheric columns can be considered accurate only above bright surfaces. Over dark surfaces such as the ocean, the sensitivity of the satellite measurements to boundary-layer BrO is unfortunately low.

*Figure 9: – please try to get onto one page, use horizontal colorbars perhaps?*

Figure 9 has been fitted to one page.

*Page 22, line 16. And Figure 11: Stating that the relationship between sea-ice area and BrO explosion is complex is unsubstantiated by a useful plot. I would like to see the first year sea-ice area versus average BrO column plot (or BrO explosion area). Figure 11d goes some of the way to providing this but I suggest that an annual number be provided i.e. those produced in 11 a and b (with variability range errorbars for both axes) could be more useful than using every data point above the threshold as is done currently. Figure 11d trend line is arbitrary and should be removed (unless the annual number provides support for a linear relationship). Some metric representation of annual cyclonic activity or amplification amplitude would be valuable too on this plot – i.e. providing an annual cyclonic index in color and some index for Arctic*

*amplification temperature with size may help to disentangle the story for us. This way we would be able to determine whether a parameterization that would be useful to test models could be derived for this 'complex' relationship.*

Removed "complex" in this line and throughout the text (see reply above), being more precise on describing the relationship between tropospheric BrO and sea ice age (for example, adding a sensitivity test). Figure 11a shows the first year sea-ice versus average BrO columns plot (as you suggested). No threshold is applied in this sub-plot. Also, we have removed the trend line from Figure 11d. We have added 1σ uncertainties for the correlation coefficients of the two scatter plots. We calculated the correlation coefficients for the annual time-series of tropospheric BrO and first year ice as well. However, we believe that the ones from the daily scatter plots (as shown in the manuscript) are more informative and describe the relationship between the two quantities more accurately. We have added the variability range error bars for both axes and show the Figure here:

[Figure]

*Page 25, line 4: VD -> VCD.*

Changed

*Figure 11: a-d labels missing.*

Added the labels

*Page 27, figure 12: How can trends over 1 year be considered significant or reliable with only a few points? I assume the annual cycle is removed for this plot (otherwise like a sine curve, you would get a trend just due to that). I'm really unclear about what figure 12 is showing, given it only is discussed briefly and doesn't add to the papers aims/conclusions. As the minimum time to detect a trend is not provided, I think this plot and the discussion can be removed.*

Figure 12 provides information on the variability of the trend of tropospheric BrO VCDs for different starting years (x-axis) and ending years (y-axis). As the value of the calculated trend depends on the averaging time period, the Figure provides a visualization of the consistency of the observed BrO increases and how they changed over time. For almost all time periods over 5 years and more, positive trends occur. As the reviewer correctly pointed out, trends over very short time periods are less significant. The annual cycle for the trends is removed by starting and ending at the same time of the year (i.e. end of March). However, as the reviewer pointed out, trends over short periods are dominated by inter-annual variability.

*Figure 13: over what time period is the trend calculated? Provide this in the caption.*

Added. The trends are calculated over a 22 year period (1996 to 2017).

*Page 29 line 5: the BrO explosion -> Bro explosion events.*

Changed as suggested

*Page 29: when discussing trends state over what period they are determined.*

The information on the time period has been added.